



# Physicochemical properties affect ice nucleating abilities of biomass burning derived charcoal aerosols at cirrus and mixed-phase cloud conditions

Fabian Mahrt[1,2], Carolin Rösch[1,3], Kunfeng Gao[1,4,5], Christopher Dreimol[6,7], Maria A. Zawadowicz[8], and Zamin A. Kanji[1]

[1]Department of Environmental System Sciences, ETH Zurich, Zurich, 8092, Switzerland
[2]now at: Laboratory of Environmental Chemistry, Paul Scherrer Institute, 5232 Villigen, Switzerland
[3]now at: City of Zurich, Environmental and Health Protection Service - Air Quality, Zurich, Switzerland
[4]School of Energy and Power Engineering, Beihang University, Beijing, China
[5]Shenyuan Honours College of Beihang University, Beihang University, Beijing, China
[6]Department of Civil, Environmental and Geomatic Engineering, Institute for Building Materials, ETH Zurich, 8093 Zurich, Switzerland
[7]Cellulose & Wood Materials Laboratory, Empa, 8600 Dübendorf, Switzerland
[8]Brookhaven National Laboratory, Environmental and Climate Sciences Department, Upton New York, USA

*Correspondence to*: Zamin A. Kanji (zamin.kanji@env.ethz.ch), Fabian Mahrt (fabian.mahrt@psi.ch)

**Abstract.** Atmospheric aerosol particles play a key role for air pollution, health and climate. Particles from biomass burning emissions are an important source of ambient aerosols and have increased over the past, and are projected to further surge in the future as a result of climate and land use changes. Particles emitted from biomass combustion are often complex mixtures of inorganic and organic materials, with soot, ash and charcoal having previously been identified as main particle types emitted. Despite their importance for climate, their ice nucleation activities remain insufficiently understood, in particular for charcoal particles, whose ice nucleation activity has not been reported. Here, we present experiments of the ice nucleation activities of 400 nm size-selected charcoal particles, derived from two different biomass fuels, namely a grass charcoal and a wood charcoal. We find that the charcoal types investigated do not contribute to ice formation via immersion freezing in mixed-phase cloud conditions. However, our results reveal a considerable heterogeneous ice nucleation activity of both charcoal types at cirrus temperatures. Inspection of the ice nucleation results together with dynamic vapor sorption measurements indicates that cirrus ice formation proceeds via pore condensation and freezing. We find wood charcoal to be more ice-active than grass charcoal at cirrus temperatures. We attribute this to the enhanced porosity and water uptake capacity of the wood compared to the grass charcoal. In support of the results, we found positive correlation of the ice nucleation activity of the wood charcoal particles and their chemical composition, specifically the presence of mineral components, based on single-particle mass spectrometry measurements. Even though correlational in nature, our results corroborate recent findings that ice-active mineral could largely govern the aerosol-cloud interactions of particles emitted from biomass burning emissions.





## 1 Introduction

Biomass burning emits greenhouse gases and aerosol particles into the atmosphere. The carbonaceous particulate matter associated with biomass burning makes up a considerable fraction of the global aerosol burden (van der Werf et al., 2010; Schill

et al., 2020; Reddy and Boucher, 2004; Andreae and Merlet, 2001). Such biomass burning aerosols are emitted from a wide range or processes, including wildfires, residential biomass combustion, and industrial combustion of biofuels. As a result of global warming, deforestation and changes in agricultural practices the frequencies and intensities of biomass burns in the form of wildfires have increased over the past and are projected to further increase, enhancing the atmospheric burden of biomass burning derived particles (Ford et al., 2018; Moritz et al., 2012; Donovan et al., 2017; Westerling et al., 2006; Marlon

et al., 2008; Holden et al., 2018). Thus, understanding the properties and environmental impacts of aerosol emitted from biomass burning is crucial. Owing to the complexity of the various emission sources that particulate matter from biomass combustion encompasses, the aerosol types can display a range of different physical and chemical properties. Major particle types emitted during biomass burning include soot, charcoal and ash particles, that are formed through different processes and at different stages of the combustion process (Chylek et al., 2015).

Soot particles form during the flaming stage of the combustion process at high temperatures, usually above 1000 °C, through condensation of gas phase intermediates (e.g., benzene, acetylene, and polycyclic aromatic hydrocarbons), i.e., by gas-to-particle conversion, leading to graphitic carbon spherules that cluster together into soot aggregates (Glassman et al., 2014). Characteristic for these soot particles is their fractal morphology, water-insolubility and microstructure, containing a high content of refractory graphitic carbon (Petzold et al., 2013). This graphitic carbon causes strong absorption of light in the near-

visible spectral region, due to the availability of nonlocalized electrons between the graphite layers. In contrast to soot, charcoal particles are formed during the smoldering stage of the combustion process, which is characterized by lower temperatures, below approximately 600 °C. At these temperatures the (organic) fuel material decomposes, usually at limited air (oxygen) access, but the carbon does not vaporize, leading to incompletely oxidized pyrolysis products, including charcoal particles (Chylek et al., 2015; Andreae and Merlet, 2001). Thus, considering their origin, charcoal particles are considered as primary

aerosol particles, constituting charred, solid biomass material. Consequently, charcoal particles often retain a recognisable structure and morphology of their respective biomass fuel source (Sharma et al., 2004; Masiello, 2004), making their morphology more variable compared to the fractal-like morphology typically observed for soot aggregates. In addition, charcoal particles are generally characterized by higher elemental ratios of oxygen-to-carbon (O/C) and hydrogen-to-carbon (H/C, Hammes et al., 2007; Kim et al., 2003) compared to soot particles, i.e., a less graphitic microstructure, conferring a reduced absorbency

(reduced sp2-bonded carbon content causes lower absorption of radiation, Bond and Bergstrom, 2006), as well as reduced thermal and oxidative stability. Another important category of particles often released during biomass burning is ash, which in contrast to soot contain only a limited amount of carbon. They denote the solid residuals from combustion organic substances, i.e., the refractory, inorganic material. These non-combustible constituents often derive from mineral inclusions or heteroatoms (excluding carbon and hydrogen) in the biomass fuel material, such as e.g., calcium (Ca), magnesium (Mn) and



iron (Fe). Previous work has suggested that up to 27 % of fire-derived carbon can result in charcoal particles (Zimmerman and Mitra, 2017). Nonetheless, an unambiguous discrimination and separation of the various particles types from biomass combustion is not always possible, and interdisciplinary accepted definitions are a matter of an ongoing debate (Andreae and Gelencser, 2006; Petzold et al., 2013). In addition, open wildfires are dynamic processes where the different stages of a combustion process are often present at the same time. Thus, the smoke plumes usually entrain combined emissions of the different

particle types. This is consistent with observations from smoke plumes, including organic and inorganic components (Maudlin et al., 2015; Vassilev et al., 2010, 2012; Posfai et al., 2003; Li et al., 2003; Adachi et al., 2022). Moreover, previous studies have also shown that biomass burning particles can contain crystalline mineral phases that form during the combustion process (Vassilev et al., 2013; Gao et al., 2014), resulting from elements that are present within the fuel, including aluminium (Al), Ca, potassium (K), Fe, sodium (Na), and silicium (Si, Currie and Perry, 2007; Vassilev et al., 2010).

It is widely recognized that biomass burning particles have an important role in many environmental compartments. For instance, aerosol particles from biomass combustion affect the carbon cycle and biogeochemical processes when incorporated into surface reservoirs (e.g., Santín et al., 2016). Biomass burning derived particles also affect visibility and contribute to poor air quality, affecting human health (Reddington et al., 2015; Chen et al., 2017; Pardo et al., 2020). Yet, biomass burning particles also play a central role in atmospheric processes (Crutzen and Andreae, 1990; Andreae et al., 1994; Sokolik et al.,

2019). For instance, they can affect the hydrological cycle and climate, directly through absorption and scattering of light, as well as indirectly when acting as cloud nuclei, affecting precipitation formation and cloud microphysical properties, with important ramifications for the regional and global energy budget (Bond et al., 2013; Bond and Bergstrom, 2006; Penner et al., 1992; Chylek and Wong, 1995; Koren et al., 2004; Kaufman and Koren, 2006; Kaufman and Fraser, 1997; Ditas et al., 2018; Yue et al., 2022). Despite the surge in emissions of biomass burning particles and their importance for climate, their

aerosol-cloud interactions, in particular their ability to act as ice nucleating particles (INPs, Vali et al., 2015) remains associated with large uncertainties (Bond et al., 2013; Sokolik et al., 2019).

Several previous field and laboratory studies have reported the ice nucleation properties of particles derived from biomass burning for conditions relevant for cirrus or mixed-phase clouds (MPC, Petters et al., 2009; Jahn et al., 2020; Jahl et al., 2021; Korhonen et al., 2020; Levin et al., 2016; McCluskey et al., 2014; Twohy et al., 2010; Chou et al., 2013; DeMott et al., 2009;

Adams et al., 2020; Barry et al., 2021). For instance, DeMott et al. (2009) performed ice nucleation experiments at cirrus temperatures, i.e., $T < 235$ K, and found that ice formation on 100 nm diameter particles emitted from combustion of a range of fuel types, including rice straw, grass and wood in most cases was indistinguishable from homogeneous freezing of aqueous solution droplets. Similarly, studies investigating the ice nucleation activity at mixed-phase cloud (MPC) conditions, i.e., $T > 235$ K and relative humidities with respect to water ($RH_w$) larger than 100 %, have often found particles emitted from biomass

combustion to have no or negligible heterogeneous immersion freezing activities within this temperature regime (e.g., Petters et al., 2009; Adams et al., 2020; Korhonen et al., 2020). Interestingly, biomass burning aerosols that formed ice via immersion freezing often contained inorganic material (McCluskey et al., 2014; Petters et al., 2009; Jahl et al., 2021; Jahn et al., 2020). For example, Petters et al. (2009) investigated the differences between the fuels that produced and did not produce INPs and



observed that the fuel types producing INPs overall had a higher probability of containing K, nitrite ($NO_2^-$) and inorganic

material, compared to the fuels that did not show ice nucleation, but only provided correlational evidence. Similarly, McCluskey et al. (2014) found that the INPs from two different wildfire were dominated by mineral or metal oxides, contributing 78 % and 67 %, respectively, to the total number of INPs analyzed per wildfire event (50-60 individual particles analyzed per event), with high abundance of Ca and Al being characteristic features of these ice active particles. Consistent with these observations, Jahn et al. (2020) recently attributed the immersion freezing ice nucleation activity to the inorganic components

present within bottom ash particles produced during biomass burning, as well as biomass burning aerosol particles sampled during combustion of grass and wood fuels. Specifically, Jahn et al. (2020) found crystalline Ca- and Si-containing mineral phases, that were both released and newly formed during the combustion process, to largely govern the ice nucleation activity of these particles. Similarly, a positive correlation between the presence of $SiO_2$ and and Ca and the immersion freezing activity of ash particles has previously been reported (Grawe et al., 2016, 2018). The overall conclusions emerging from these studies

is that inorganic, mineral components are likely important for the ice nucleation activity of biomass burning particles. Nonetheless, due to the variety of different particle types emitted from biomass burning and the range of associated physicochemical properties, considerable uncertainty still exists regarding the ice nucleation activity of biomass burning particles. One particle type that is often emitted during biomass burning, but whose ice nucleation activity has received little attention to date are charcoal particles.

In this study, we investigate the ice nucleation ability of charcoal particles at temperatures relevant for both cirrus and MPC regimes in laboratory experiments. We studied submicron charcoal particles from two different types of biomasses, from wood and grass fuel, in an attempt to cover different emission sources relevant for atmospheric charcoal particles. A suite of auxiliary measurements, including particle water uptake and loss, morphology and chemical composition, complement our ice nucleation data to characterize the particle physicochemical properties and explore potential links to their ice nucleation activity.

## 2 Materials and Methods

### 2.1 Charcoal samples

Charcoal materials were obtained from the University of Zurich, Switzerland. The samples investigated are from two types biomass fuel types, including a wood and a grass charcoal. The wood charcoal was sourced from debarked wood of chestnut trees (Castanea sativa) while the grass charcoal originated from rice straw (Oryza sativa). Charcoal particles from chestnut

trees were used as surrogates for aerosols emitted from biomass burning of larger tree species. The Castanea sativa studied here is representative for forest growth across Europe, where it covers approximately 2.25 million hectares (Conedera et al., 2004). By contrast, charcoal particles from rice straw were used as surrogates for aerosols emitted from biomass burning in agricultural fields. Rice is one of the most important crops globally and burning of its straw is commonly practiced in many regions around the world to remove residues after harvest or used as a low-cost fuel in heating stoves (Singh et al., 2021).






Both types of charcoal particles were produced by pyrolysis in a pure nitrogen atmosphere at a temperature of 450 °C using a commercial furnace (Carbolite Gero GmbH & Co., Neuhausen, Germany, Model: GERO GLO 11/40), operated at a nitrogen flow rate of 500 L h$^{-1}$. The charcoal particles investigated here represent typical low temperature charcoals (Hammes et al., 2006). Details on the production method and a description of the physical and chemical properties is given elsewhere (Hammes

et al., 2006, 2008). In brief, the O/C and H/C elemental ratios were found to be identical for wood charcoal and grass charcoal and were reported to be 0.3 and 0.7, respectively. In addition, both samples were dominated by aryl functionalities, i.e., aromatic carbon (70 %), as found by nuclear magnetic resonance spectroscopy, indicating overall low abundance of hydrogen. The specific surface areas were found to be 2.0 m$^2$ g$^{-1}$ ± 10 % and 5.9 m$^2$ g$^{-1}$ ± 10 %, for the wood and grass charcoal particles, respectively, determined by nitrogen (N$_2$) adsorption following the Brunauer–Emmett–Teller (BET) method (Brunauer et al.,

1938). Hence, the surface areas of the charcoal particles investigated here are about an order of magnitude lower compared to other carbon-rich materials such as e.g., soot (Ouf et al., 2019). We note that these specific surface area values are characteristic for low temperature charcoals and that it has previously been shown that the properties of charcoal particles can strongly depend on the combustion conditions. For instance, the specific surface area of charcoal samples has been shown to increase with increasing maximum pyrolysis temperature, reaching values of several hundred square meters per gram of particle mate-

rial for pyrolysis temperatures higher than those used here (Brown et al., 2006).

For the ice nucleation experiments, the charcoal samples were aerosolized using a rotating brush aerosol generator (RBG, Palas, Model 1000) and passed through a cyclone (URG Corporation., Model 2000-30EHB, 50 % cut-off diameter ~1 μm at a flow rate of 16.7 L min$^{-1}$) in order to confine the aerosol size distribution to mostly submicron particles, before entering a 2.7 m$^3$ stainless steel tank (Kanji et al., 2013). After filling the tank with charcoal aerosol up to a number concentration of approx-

imately 4000–5000 cm$^{-3}$ the particles were kept suspended with a continuously stirring fan mounted at the bottom of the tank. Particles for the ice nucleation experiments and for aerosol analysis were directly sampled from the tank and a make up flow of particle-free, high purity nitrogen was used to keep the pressure in the tank constant throughout an experiment.

## 2.2 Ice nucleation measurements

Ice nucleation experiments were performed on size-selected aerosol particles at conditions relevant for both cirrus and MPCs.

An aerosol stream sampled from the stainless-steel tank was passed through a differential mobility analyzer (DMA, TSI Inc., Model 3080, polonium source aerosol-to-sheath flow ratio of 1:7) to select a quasi-monodisperse aerosol population with mobility diameters ($d_m$) of $d_m$ = 400 nm. The fraction of double-charged particles at the used flow conditions was approximately 15 % (dm = 699 nm, Wiedensohler, 1988).

### 2.2.1 HINC measurements (cirrus temperature regime)

Ice nucleation experiments in the cirrus temperature regime were performed using the horizontal ice nucleation chamber (HINC), and is described in detail elsewhere (Lacher et al., 2017). Briefly, in HINC two ice-coated, horizontally oriented copper plates are individually cooled to different temperatures, with uncertainties of 0.1 K, so that a linear temperature gradient





is established across the vertical extent of the ice chamber. This results in a water vapor supersaturation profile across the chamber, due to the non-linear relation between temperature and saturation water vapor pressure, described by the Clausius-Clapeyron equation (e.g., Lohmann et al., 2016). The aerosol particles are assumed to be mostly transported through the CFDC within the so-called aerosol lamina at the center of the chamber. The thickness (vertical extent) of the aerosol lamina is constrained by the ratio of a particle-free nitrogen sheath flow and the aerosol sample flow. For the experiments presented here aerosol-to-sheath flow ratios between approximately 1:10 and 1:12 were used. The vertical extent of the aerosol lamina decreases with decreasing values for the aerosol-to-sheath flow ratio, ultimately causing the variation of $T$ and RH the charcoal particles are exposed to within the aerosol lamina to decrease. The RH-conditions across the aerosol lamina at the center of the chamber are calculated from the linear temperature profile between the two horizontal chamber walls, the parametrization given by Murphy and Koop (2005), and conservatively assuming a lower limit for the aerosol-to-sheath flow ratio of 1:10 (see Mahrt et al., 2018 for details). This translates to a maximum uncertainty of the relative humidity with respect to water (RH$_w$; and with respect to ice, RH$_i$) of approximately $\pm 3$ % (5 %) at $T = 218$ K. The particle residence time ($\tau$) within HINC can be varied by adjusting the position of a movable aerosol injector. However, the position was fixed for all our experiments corresponding to an average residence time of approximately 10 s for the temperature range investigated here. Ice nucleation experiments in HINC were conducted as RH-scans, where the RH is slowly increased ($< 3$ % RH$_i$ min$^{-1}$) at a fixed center temperature, by increasing the temperature gradient between the two walls. The number of aerosol particles that act as INPs and form macroscopic ice crystal ($n_{ice}$) at a given RH are detected at the exit of HINC. By dividing $n_{ice}$ by the total number concentration of aerosol particles entering HINC, $n_{tot}$, we calculate the activated fraction ($AF$) of charcoal particles given as:

$$AF = \frac{n_{ice}}{n_{tot}}.$$  (1)

For all HINC experiments reported here, $n_{ice}$ is given by the number of particles with diameters larger than 1 μm, as detected by an optical particle counter (OPC, Model GT-526S, MetOne Instruments, Grants Pass, Oregon, USA) at the exit of HINC. The value of $n_{tot}$ was determined by a condensation particle counter (CPC, TSI Inc., Model 3776) operated in low flow mode (0.3 L min$^{-1}$), in parallel to HINC, and taking the dilution factor due to the aerosol-to-sheath flow into account. Uncertainty in the $AF$ is 14 %, resulting from the propagated counting uncertainties of 10 % from each the OPC and the CPC. Each $AF$ curve is corrected for background counts from e.g., chamber-internal frost particles, by measuring the $AF$ for particle-filtered air at the start and target RH of each RH-scan for a period of 5 min and linearly interpolating the background OPC counts for the RH range in-between. All $AF$ curves reported are averages of two to three independent RH-scans. In HINC ice nucleation experiments were performed at $218 \leq T \leq 233$ K, which are relevant temperatures for cirrus cloud formation.

## 2.2.2 IMCA-ZINC measurements (MPC temperature regime)

Immersion freezing mode experiments at $T > 230$ K were performed using a combination of the Zurich ice nucleation chamber (ZINC, Stetzer et al., 2008; Welti et al., 2009) and its vertical extension, the immersion mode cooling chamber (IMCA, Lüönd





et al., 2010). The experimental setup has been described in detail previously (Lüönd et al., 2010; Welti et al., 2012). In short,
particles are first activated into cloud droplets in IMCA at $T = 313$ K. High supersaturation with respect to water of approximately 20 % at the top part of IMCA ensures cloud droplet activation of all particles, so that each aerosol particle becomes immersed into a single cloud droplet. As the particles flow through IMCA, the droplets with the charcoal particles immersed are continuously cooled to the experimental (aerosol lamina) temperature of ZINC and lowering the supersaturation to a value slightly above water saturation, in order to avoid evaporation of the droplets upon reaching the supercooled regions of ZINC. In ZINC, a vertical CFDC with parallel-plate design, the droplets are exposed to freezing temperatures between approximately 230 K to 255 K. Immersion freezing experiments were performed using so-called *T*-scans, where the $RH_w$ is fixed to value of 104 % and the temperature is decreased in 2 K increments, from low to high values. At each temperature increment the *AF* is used to determine the fraction of frozen droplets at the end of ZINC, given by:

$$AF_{IMCA-ZINC} = \frac{n_{ice}}{n_{ice}+n_{droplets}}. \tag{2}$$

Here, $n_{ice}$ and $n_{droplets}$ define the number concentration of ice crystals and cloud droplets, respectively, formed within IMCA-ZINC and detected with an OPC (Lighthouse, Model: Remote 5104) at the exit of the ZINC chamber. The uncertainty in *AF* results from a counting uncertainty of 10 % from each the OPC and the CPC resulting in a 14 % relative error. The initial particle number concentration entering IMCA-ZINC was recorded by a CPC (TSI Inc., Model 3787) operated in parallel. In the case of IMCA-ZINC experiments, $n_{ice}$ denotes the number of particles with $d_p > 2$ μm, as determined by the OPC. All *AF* curves are an average of three individual experimental runs and corrected for background counts determined from sampling particle free air for a duration of 3 min at each temperature of the *T*-scan. The temperature uncertainty in IMCA-ZINC measurements was ± 0.1 K, translating into a maximum uncertainty in $RH_w$ ($RH_i$) of approximately 3 % (5 %) at 218 K. The average particle residence time within ZINC was τ ≈ 10 s over the temperature range investigated here.

## 2.3 Auxiliary measurements for sample characterization

### 2.3.1 Transmission electron microscopy measurements

Aerosol particles from each sample were collected for TEM analysis. Polydisperse aerosol particles were sampled from the stainless-steel tank and impacted on standard Cu-TEM grids having a continuous carbon film and 400 mesh (Quantifoil Mirco Tools GmbH, Großlöbichau, Germany). Particles were collected using the Zurich electron microscope impactor (ZEMI, Mahrt et al., 2018), a rotating drum impactor allowing for individual control of sample flow rate, sample time and impaction distance. Collected particles were analysed using a TEM JEOL-1400+ microscope (JEOL Ltd., Tokyo, Japan) equipped with a LaB6 filament, and operating at 120 kV.





### 2.3.2 Water vapor sorption measurements

Particle hygroscopicity was characterized by dynamic vapor sorption (DVS, Model Advantage ET 1, Surface Measurement

Systems Ltd., London, UK). DVS gravimetrically measures the amount of water being taken up by the bulk sample, by deter-

225    mining the relative mass change, $\Delta m$, between the sample under dry conditions ($RH_w = 0$ %) and at each $RH_w$ value during a

DVS-scan. The DVS instrument has a weighting precision of 0.1 µg and the absolute uncertainty in $RH_w$ amounts to $\pm$ 0.5 %.

The flow rate through the DVS was 200 ml min$^{-1}$ of high purity $N_2$ (5.0 grade) for our experiments, and the RH was controlled

by humidifying the $N_2$-flow. The bulk sample water uptake and loss were determined at 298 K by measuring the adsorption

and desorption isotherms in 5 % and 3 % $RH_w$ steps in the range of 0–80 % and 80–98 %, respectively. The value of $\Delta m$ was

230    determined at each $RH_w$ step after quasi-equilibrium was reached, which was defined as a mass change rate below 0.0005 %

min$^{-1}$ over a period of 10 min. If this criterion was not met, the mass that was reached after a maximum period of 1000 min at

a given $RH_w$ step, was used instead to calculate the amount of water that was taken up during quasi-equilibrium. Prior to the

DVS analysis each sample was dried directly within the DVS sample chamber at a temperature of 298 K and over a period of

1000 min, in order to outgas any pre-adsorbed water, followed by a full sorption cycle, starting with the adsorption and sub-

sequently the desorption measurements.

### 2.3.3 Chemical characterization

The chemical composition of the particles was analyzed using single-particle aerosol mass spectrometry. Specifically, we

employed an aerosol time of flight mass spectrometer (ATOFMS, Model 3800; TSI Inc.) to obtain the chemical composition

of individual aerosol particles as a function of particle size. A detailed description of the ATOFMS can be found elsewhere

(Gard et al., 1997; Noble and Prather, 1996; Prather et al., 1994). In brief, aerosols are sampled into the instrument through a

critical orifice. Downstream of the orifice an aerodynamic focusing lens (AFL), inertially focusses the particles, which exit the

AFL through an acceleration nozzle, controlling the supersonic expansion of the particle loaded air and causing a size depend-

ent acceleration of the aerosols. Particle velocity is determined from light scattering pulses by measuring the transit time

between two diode-pumped solid-state continuous wave laser beams operating at a wavelength of $\lambda = 405$ nm (Livermore

Instrument Inc., Model FMXL405). The vacuum aerodynamic particle diameter, $d_{va}$, is calculated from the measured velocity

using a calibration curve obtained when using polystyrene latex spheres of known size and density. In the mass spectrometry

source region a pulsed ultra violet Neodynium: yittrium-aluminium-garnett (Nd:YAG) laser (Big Sky Laser Technologies Inc.,

Model Ultra Series 230) with a wavelength of $\lambda = 266$ nm is used to ionize particles. The ATOFMS employs a laser ablation

ionization technique, which allows to detect refractory aerosol components within the charcoal particles. The beam diameter

of the Nd:YAG laser used was ~2.4 mm and the energy of the pulse can be varied, enabling some control over the fragmentation

of the analyte. Here, a typical pulse energy of about 1 mJ was used in order to allow for a more complete ionization of the

particles. Positive and negative ions are detected simultaneously using a dual-reflectron time-of-flight mass spectrometer.



The mass calibration of the time-of-flight signal was obtained from analyzing particles of known chemical composition prior
to the experiments. Single particle mass spectra from the ATOFMS were imported into the TSI firmware MS-Analyze, where
basic analysis, i.e., mass and size calibrations were applied to the data sets. For evaluation of the single particle mass spectral
data, the calibrated data sets were then imported and processed within the flexible analysis toolkit for the exploration of single-
particle mass spectrometer data (FATES, Sultana et al., 2017). FATES includes different features for ATOFMS data postpro-
cessing, such as sorting the mass into groups, or converting the mass spectra into spectra with unit mass resolution ("stick
spectra"), which is achieved by integrating the ion signal peak width around integer mass-to-charge ($m/z$) values, or calculating
average spectra for a given set of single particle spectra. Here we report average mass spectra in terms of relative peak area
and focus on the analysis of the averaged stick spectra of each charcoal type, examining these with respect to the presence of
marker peaks. We note that different ions can result in the same $m/z$ ratio and here peak assignment is informed by comparison
to literature data, where different particles types of known composition with instruments that use a $\lambda = 266$ nm Nd:YAG laser
for aerosol particle ablation and ionization, including other (commercial) ATOFMS instruments, as well as the ALABAMA
and the SPASS mass spectrometers (Brands et al., 2011; Erdmann et al., 2005). The particle analysis by the ATOFMS was
achieved directly sampling from the stainless-steel aerosol tank, similar to the ice nucleation measurements, but using poly-
disperse aerosol. A total of 11'135 and 12'357 single-particle mass spectra were obtained for grass charcoal and wood charcoal,
respectively.

## 3. Results and discussion

### 3.1 Ice nucleation ability

Figure 1 shows the $AF$ of wood and grass charcoal as a function of temperature (230–255 K) relevant for MPC conditions, as
obtained from the IMCA-ZINC experiments. The 400 nm diameter wood charcoal and grass charcoal particles showed a
similar ice nucleation activity in this range, with the $AF$-curves overlapping for both samples within experimental uncertainties,
overall revealing a low immersion freezing ability. Specifically, values of $AF \leq 0.1$ were observed at $T > 237$ K for both wood
charcoal and grass charcoal particles. At $T < 237$ K the $AF$ increased, reaching values of around 0.59 and 0.66 for wood
charcoal and grass charcoal particles, respectively, at $T = 235$ K, i.e., at the homogeneous ice nucleation temperature (HNT)
of water. We indicate homogeneous freezing condition in Fig. 1 by the gray shading, using the parametrization of Ickes et al.
(2015). As the temperature in ZINC was further decreased below the HNT, the $AF$ for both charcoal types increased, reaching
maximum values of up to ~0.96 at $T = 231$ K. We note that particles leaving the aerosol lamina within ZINC, outside which
they are exposed to different conditions of $T$ and RH, can cause $AF$ values below unity (DeMott et al., 2015; Garimella et al.,
2017). The sharp increase in $AF$ below temperatures of 237 K likely results from homogeneous freezing of soluble organic or
inorganic material associated with the charcoal particles. Given that the maximum temperature during pyrolysis of the initial
biomass material was 450 °C, most of the organics should have vaporized (Stratakis and Stamatelos, 2003) and carried away
with the $N_2$ purge flow during pyrolysis. Hence, the increase in $AF$ observed in our immersion freezing experiments is likely


caused by soluble inorganic material. This is further corroborated by the mass spectral signatures obtained from our ATOFMS sampling discussed below. Overall, we conclude from our experiments shown in Fig. 1 that the charcoal particles investigated here are poor immersion mode INPs. At first sight, our results appear in contrast to those of earlier studies that reported appreciable immersion freezing activity of biomass burning particles, as discussed above. One way to reconcile the results reported here with those of previous studies is considering the particle surface area per droplet used to study the immersion

freezing activity of biomass burning particles, that catalyzes their heterogeneous freezing activities (Lohmann et al., 2016). By accounting for the particles surface area, the $AF$ values observed in our experiments can be normalized to derive ice active surface site densities, $n_s$, (e.g., Vali, 2014), that can directly be compared to those of previous studies. Assuming spherical particle geometry for our particles, one 400 nm charcoal particle immersed in a single droplet has a surface area of ~0.5 μm². This likely denotes an underestimation of the (true) particle surface area due to the complex morphology of the charcoal

aerosols. Relying on this idealized assumption and using the $N_2$-based BET specific surface area of the charcoal types reported in Hammes et al. (2006) (2.0 m² g⁻¹ for wood charcoal, 5.9 m² g⁻¹ for grass charcoal), together with typical charcoal particle densities of 0.75 g cm⁻³ (Sander and Gee, 1990), we estimate surface areas of ~50 μm² and 140 μm² for a 400 nm diameter wood and grass charcoal particle, respectively. This corresponds to $n_s$ values of ~10 to 80 cm⁻² at $T$ = 244 K. By contrast, previous studies showing biomass burning particles to be efficient immersion freezing nuclei were mostly performed on pol-

ydisperse combustion aerosol populations or by using droplets in cold stage techniques. In such cases, each droplet can contain multiple biomass burning particles. Thus, the particle (or INP) surface area per droplet is larger in such cases, increasing the freezing probability compared to our measurements. As an example, Jahn et al. (2020) reported $n_s$ values of approximately 8 10³ cm⁻² for their most ice-active ash sample tested and at the lowest temperature investigated of $T$ = 244 K (see their Fig. 1b), which is orders of magnitude higher than the $n_s$ values observed here at the same temperature. This comparison illustrates that

the surface area of our charcoal particles per droplet was likely insufficient to catalyze immersion freezing. Therefore, we conclude that biomass burning particles with properties (composition and surface area) similar to the charcoal types studied here can likely not compete with the immersion freezing abilities of other atmospheric aerosol particles, such as mineral dust (e.g., Murray et al., 2012).

    Although the 400 nm diameter charcoal particles showed negligible immersion freezing activity, they revealed an enhanced

ice nucleation ability at temperatures below the HNT. In Fig. 2 we show the complete $AF$-curves for the different cirrus temperatures investigated here, as obtained from our RH-scans performed in HINC. At $T$ < 233 K the charcoal particles showed a considerable ice nucleation activity for RH < RH_hom, where RH_hom denotes conditions for homogeneous freezing of aqueous solution droplets, indicated by the black dashed lines (Fig. 2). In addition, we observed different ice nucleation activities between the grass charcoal and the wood charcoal particles, with the latter being overall more ice active at the cirrus temper-

atures investigated. As an illustration, the ice nucleation onset (corresponding to conditions where a value of $AF$ = 10⁻² is reached) at 218 K was as low as RH_w = 78 % (130 % RH_i) for the wood charcoal particles but for the grass charcoal particles the onset was observed around and 83 % (138.5 % RH_i). We discuss these observed differences in more detail deriving support from the auxiliary measurements.



As far as the ice nucleation activity of the charcoal particles in the cirrus temperature regime is concerned, we interpret this to
result from a pore condensation freezing mechanism (PCF, Christenson, 2013; David et al., 2019; Fukuta, 1966; Higuchi and
Fukuta, 1966; Koop, 2017; Marcolli, 2014). During PCF, water is taken up into pores and cracks available on the charcoal
particles below water saturation, by capillary condensation, due to the lower equilibrium vapor pressure in confined spaces,
compared to bulk liquid water (Kelvin effect, e.g., Marcolli, 2014; Fisher et al., 1981). At $T \leq$ HNT this pore water can freeze
homogeneously. Once formed, the microscopic pore ice can grow out of the pore to form macroscopic ice crystals (David et
al., 2019; Koop, 2017), which can be detected in our cirrus temperature experiments with the OPC. An ice formation mecha-
nism on charcoal via PCF is consistent with recent studies, reporting PCF as the dominant ice formation mechanism for other
combustion particle types including ash (Umo et al., 2019; Kilchhofer et al., 2021) and soot (Mahrt et al., 2018; Nichman et
al., 2019; Jantsch and Koop, 2021; Marcolli et al., 2021; Gao et al., 2022). Ice formation via PCF is strongly determined by
the aerosol physicochemical properties (Marcolli et al., 2021; David et al., 2020). For instance, the study by David et al. (2020)
showed that during PCF pore filling with liquid water and ice growth out of the pore is largely controlled by the pore size and
the contact angle between water/ice and the particle surface (making up the pore wall material), when investigating different
synthetic silica particles with well-defined pore diameter and varying their surface functional groups to mimic different contact
angles. The absence of ice formation below water saturation for $T >$ HNT, as confirmed by additional RH-scans performed
with HINC at temperatures of 243 K and 238 K (see Fig. A1), further suggests that the charcoal particles investigated herein
form ice via PCF and lack the active sites for immersion freezing. The marked dependence of ice formation on HNT is incom-
patible with the classical deposition ice nucleation (Welti et al., 2014) and provides strong evidence that ice formation on the
charcoal particles investigated here occurs via liquid water in support of PCF. We further point out that the absence of ice
nucleation for RH < $RH_{hom}$ at $T = 233$ K is not inconsistent with PCF, but is explained by accounting for the negative pressure
(tension of the water meniscus) experienced by the capillary condensate, i.e., the supercooled pore water (Marcolli, 2020).
Marcolli (2020) demonstrated that at temperatures close to the HNT capillary condensates within pores only form close to
water saturation. At these humidity conditions, the pore water experiences almost ambient pressure conditions, i.e., hardly any
negative pressure. In turn, this causes the homogeneous ice nucleation rates to be lower compared to the cases at 228 K, 223
K and 218 K, where we observed ice nucleation on the charcoal particles at considerably lower relative humidities and below
homogeneous freezing conditions (see Fig. 2). At these temperatures the charcoal pores fill at lower RH compared to the case
of $T = 233$ K, hence the pore water experiences higher negative pressures (Marcolli, 2020) and therefore higher homogeneous
ice nucleation rates. We thus conclude that at $T = 233$ K the homogeneous ice nucleation rates are too low to observe ice
formation within the particle residence time of ~10s within HINC (David et al., 2019; Marcolli, 2020).

Altogether, the high ice supersaturations required to nucleate ice on these particles at $T = 233$ K, and the absence of heteroge-
neous ice formation at $T > 233$ K (Fig. A1 and Fig. 1), render it unlikely that ambient aerosols with physicochemical properties
similar to the charcoal particles investigated here contribute to heterogeneous ice nucleation at $T >$ HNT.





### 3.2 Water vapor sorption

Figure 3a shows the water vapor sorption and desorption isotherms of the two charcoal samples, determined by DVS experiments. These measurements allow us to simultaneously infer information of the charcoal hydrophilicity and porosity. During DVS water uptake and loss is determined gravimetrically and in Fig. 3a we show the mass change, $\Delta m$, denoting the quasi

equilibrated (steady state) moist sample mass at each $RH_w$ value, relative to the dry sample mass at 0 % $RH_w$. The solid lines and filled symbols in Fig. 3a represent water uptake isotherms and the dashed lines and open symbols represent the water loss isotherms for the same DVS run.

The grass charcoal sample exhibits a shallow water uptake curve up to 80 % $RH_w$, with $\Delta m < 0.5$ %. The corresponding water vapor-based BET specific surface area of the grass charcoal sample was determined as $BET_{H2O} = 7.9$ g m$^{-2}$ (see Appendix D).

Between 83 % to 98 % $RH_w$ the grass charcoal shows a more pronounced water uptake with a maximum in $\Delta m$ of around 3 % being reached at the largest $RH_w$ value (98 % $RH_w$). The absence of a plateau at these high $RH_w$ suggests that formation of a water multilayer and free water would occur if the $RH_w$ was further increased. We interpret the gradual curvature of the water uptake observed for the grass charcoal as a type II isotherm, following the IUPAC (International Union of Pure and Applied Chemistry) classification recommendations (Thommes et al., 2015; Sing et al., 1985). While type II isotherms are characteristic

for macroporous (pore diameter > 50 nm) adsorbents, the abrupt change in water uptake around 83 % and the enhanced water uptake at higher $RH_w$ indicates some water uptake by filling of mesopores (pore diameter between 2 nm to 50 nm, Thommes et al., 2015). The presence of mesopores on the grass charcoal is further supported by the (weak) hysteresis observed between the water uptake and the water loss curves and is consistent with (weak) PCF ice formation observed for the grass charcoal particles at some cirrus temperatures (Fig. 2, $T \leq 223$ K). The hysteresis of the grass charcoal is typical for a H3-type hysteresis,

which is often observed for adsorbents with a type II adsorption isotherm (Thommes et al., 2015). Nonetheless, we note that overall, the observed hysteresis is weak. This indicates that while some mesopores are present on the grass charcoal particles, this sample is likely dominated by macropores. A H3-type hysteresis loop and a dominance of macropores is characteristic for aggregates of plate-like particles (Thommes et al., 2015). A plate-like particle morphology for the charcoal aerosols is qualitatively supported by our TEM images shown in Fig. 3b (see also Fig. C1), and is further consistent with previous studies

reporting that charcoal particles mostly sustain the shape of the fuel material to some degree (Sharma et al., 2004).

The wood charcoal sample shows a distinctively different water uptake and loss behavior compared to the grass charcoal sample. Specifically, the water uptake curve of the wood charcoal reveals a pronounced concave curvature with an initial steep increase reaching values of $\Delta m = 6.5$ % at 40 % $RH_w$, and a more gradual uptake as the $RH_w$ is further increased. We attribute this to a type IV(a) isotherm (IUPAC classification recommendation), although we note that a final saturation plateau at high

$RH_w$ was not observed for the wood charcoal sample. The strong increase in sample mass at very low $RH_w$ is a common feature of samples that have micropores (pore diameter < 2 nm Thommes et al., 2015) and/or large specific surface area. The $H_2O$-based BET specific surface area of the wood charcoal sample corresponding to the water uptake isotherm of our DVS meas-



urement was determined as $BET_{H2O} = 161.3$ g m$^{-2}$ (see Appendix D). A similar adsorption behavior has previously been re-
ported for microporous combustion particles (e.g., Popovitcheva et al., 2000; Popovicheva et al., 2008). The relatively broad

$RH_w$ range over which the concave shape of the water uptake isotherm extends in our experiment, suggests a broad range of
differently sized micropores. We note that such micropores are negligible for ice formation via PCF, because micropores are
too narrow to accommodate a critical ice embryo needed for ice formation. Nonetheless, our DVS measurements also contain
characteristic features that indicates the presence of mesopores. The wood charcoal DVS measurements are accompanied by
a clear hysteresis, typical for type IV(a) isotherms. We attribute this hysteresis to a H4-type hysteresis that is typical for micro-

and mesoporous carbonaceous material (Thommes et al., 2015). The availability of mesopores on the wood charcoal sample
supports our interpretation of ice formation via PCF on these particles, described above.

Overall, our DVS measurements reveal that the wood charcoal samples show a higher water uptake capacity compared to the
grass charcoal sample. For instance, at 98 % $RH_w$, the water uptake of wood charcoal is approximately a factor ~4 larger
compared to the grass charcoal sample. We interpret the higher water affinity of the wood charcoal sample compared to the

grass charcoal to result partly from a lower charcoal-water contact angle of the former, which is consistent with the larger ice
nucleation activity of the wood charcoal particles at $T < $ HNT (see Fig. 2). However, we point out that the difference in (PCF-
) ice nucleation activity observed between the wood charcoal and grass charcoal is not constant across the cirrus temperatures
investigated here, indicating that it is not directly proportional to the hydrophilicity and mesopore volume observed in the DVS
measurements. This is consistent with previous DVS measurements and PCF-ice nucleation observations for soot particles

(Mahrt et al., 2020), and suggests that other aerosol properties such as the chemical composition might need to be considered
in order to explain the observed ice nucleation activity.

### 3.3 Chemical composition

Figure 4 shows a summary of the chemical composition of the charcoal particles, as obtained from our ATOFMS sampling.
Depicted are the average anion and cation stick spectra of grass charcoal and wood charcoal, respectively, where the peaks

denote average ion intensities in terms of relative peak area overall particles that resulted in a detectable ion signal upon being
hit with the Nd:YAG laser. Major peaks with relative area signals larger or equal to 0.02 are labelled for clarity. We point out
that the ATOFMS is a semi-quantitative instrument and that the relative peak intensities do not correspond to the actual mass
fraction of the respective component.

The anion spectrum of grass charcoal (Fig. 4a) is marked by peaks of organic ions, including the typical $C_n$ fragmentation ($C_1^-$

-$C_5^-$, $m/z$ 12…60), as well as peaks occurring at 25 ($C_2H^-$), 26 ($C_2H_2^-$/$CN^-$), 27 ($C_2H_3^-$/$NCH^-$), 49 ($C_4H^-$), and 50 ($C_4H_2^-$). The
presence of a peak at 46 ($NO_2^-$), even though at very low relative intensity, renders the presence of nitrogen containing cations
at $m/z$ of 26, and 27 likely, and is further supported by the peak at 42 ($CNO^-$). The small peak at 97 ($HSO_4^-$; low relative
intensity $<< 0.1$) suggests the presence of further secondary material on the grass charcoal particles. Additionally, the grass
charcoal contains a small peak at 79 ($PO_3^-$). Overall, these peaks are all typical marker ions occurring in aerosol particles from

biomass burning emissions (Schmidt et al., 2017; Zawadowicz et al., 2017).





Characteristic marker peaks for biomass burning aerosol are also found in the average cation spectrum of the grass charcoal (Fig. 4d), including peaks at 23 ($Na^+$), 24 ($C_2^+$) and 40 ($Ca^+/NaOH^+$). Yet, another ion typically occurring in biomass burning aerosol is potassium ($K^+$, $m/z$ 39), whose peak is dominating the average cation spectrum of grass charcoal. This is consistent with previous studies showing that mass spectra obtained by ablation and ionization mass spectrometers are often dominated

by ions with low ionization efficiencies such as alkali metals (Zelenyuk and Imre, 2005). In addition, the ionization efficiency can be impacted by the particle matrix in laser ablation mass spectrometry (Gross et al., 2000; Reilly et al., 2000), as well as the particle size and morphology (Kane and Johnston, 2000). We further note that non-uniform ablation and ionization, e.g., due to the Gaussian intensity profile of the Nd:YAG laser beam, can cause the resulting mass spectrum to not contain information on the entire particle composition (Ge et al., 1998), making the ATOMFS analysis a semi-quantitative method, as noted

above. In Fig. B1 we show the mass spectra depicted in Fig. 4 in a complementary way, to better reflect the variation in relative peak area across the entire charcoal particle population. Specifically, Fig. B1 shows what fraction of the investigated particles contained a certain ion species, together with the relative ion signal intensity. Even though there is some variation in the relative peak intensities, we note that a considerable fraction of the particles contains the characteristic (average) ion peaks shown in Fig. 4. The number size distributions of the particles contributing to the average spectra in Figs. B1 and 4 are shown in Fig.

B2. For both charcoal types, the mode of the particle size distribution is in the submicrometer range, between 300 to 500 nm (vacuum aerodynamic diameter). Thus, we consider the average spectra shown in Fig. 4 to be representative for the 400 nm charcoal particles used for our ice nucleation experiments.

The average anion spectrum of the wood charcoal is shown in Fig. 4b. The wood charcoal particles contained typical marker ions at $C_1^--C_6^-$ ($m/z$ 12…72) and associated hydrocarbon peaks at 25 ($C_2^-$), 26 ($C_2H_2^-$), 37 ($C_3H^-$), 49 ($C_4H^-$), 50 ($C_4H_2^-$), 51

($C_4H_3^-$), 52 ($C_4H_4^-$), 73 ($C_6H^-$) and 74 ($C_6H_2^-$), although some of these peaks have relative areas below 0.1 (see Fig. B1c). In addition, peaks of secondary material were also present at 97 ($HSO_4^-$). A peak that was present in the anion spectrum of wood but not in grass charcoal appears at $m/z$ of 76, which we attribute to $AlO_2(OH)^-$. Alumina species are known to contribute to ice nucleation activity of ambient particles, as they form significant components of atmospherically relevant mineral particles such as illite, which are also known to be porous, ice-active aerosol particles (David et al., 2019; Hiranuma et al., 2015; Welti

et al., 2009).

The presence of aluminium is further corroborated by the average cation spectrum of wood charcoal depicted in Fig. 4e, showing a prominent peak at 27 ($Al^+$). Next to aluminium the cation wood charcoal spectrum indicates the presence of other metals ions such as 56 ($Fe^+$). Also present are peaks at 23 ($Na^+$), 24 ($Mg^+$) and 40 ($Ca^+$). These peaks are characteristic for mineral material, and have previously been reported for e.g. dust samples (e.g., Gallavardin et al., 2008; Marsden et al., 2019).

In addition, the cation spectrum of wood charcoal is characterized by the presence of 39 ($K^+$), as well as a $C_n$ fragmentation pattern ($C_1^+-C_3^+$, $m/z$ 12, 24, 26). Overall, the peaks observed in the anion and cation spectrum of wood charcoal contain ions often observed in spectra of fly ash particles, as reported previously (Zawadowicz et al., 2017; Grawe et al., 2018; Xu et al., 2018). For example, similar peaks have previously been reported by Grawe et al. (2018) when sampling aerosol particles from





different ash types sourced from coal-burning power plants and also measured with a single particle mass spectrometer, where
more than 80 % of all particles samples (across all ash types) contained peaks at 27 ($Al^+$) and 40 ($Ca^+$).

## 4. Summary and Conclusions

In this study we investigated the ice nucleation ability of charcoal particles, an important class of aerosols generated during biomass combustion. Two charcoal samples were tested, derived from relevant biomass types, namely wood and grass charcoal. Ice nucleation experiments were carried out on 400 nm size-selected particles. Such submicron charcoal particles ex-
pected to be transported into higher altitudes (Clark and Hussey, 1996; Gilgen et al., 2018) where they can affect cloud formation. Ice nucleation activities were tested over a temperature range of 218 K $\leq T \leq$ 253 K. In the MPC regime ($T >$ 233 K), the ice nucleation was tested in immersion freezing mode, while for the cirrus temperatures ($T \leq$ 233 K), RH-scans were performed, covering a range of $RH_w$ ($RH_i$) 65–105 % (100–180 %). The bulk aerosol water uptake and loss were characterized gravimetrically by means of dynamic water vapor sorption (DVS) measurements, allowing to infer material porosity, and the
chemical composition of the particles was characterized using a single particle aerosol time-of-flight mass spectrometer (ATOFMS).

We found that the 400 nm charcoal particles investigated here were poor immersion freezing INPs at MPC temperatures, independent of the charcoal type, even in the presence of mineral components. In comparison to previous studies that often reported some immersion freezing of biomass burning particles, we attributed this to the small particle (INP) surface area
available per droplet in our experiments. By contrast, at cirrus temperatures, we found a considerable ice nucleation ability of the charcoal particles at relative humidities below those required for homogeneous freezing of solution droplets. This dependence of the ice formation ability on the homogeneous ice nucleation temperature of pure water (HNT = 235 K) provides ample evidence that ice nucleation on charcoal proceeds via a pore condensation and freezing (PCF) mechanism. At $T <$ 233 K, the observed ice nucleation activity was higher for wood charcoal compared to grass charcoal particles. In this temperature range,
the ice nucleation activity correlated with both physical and chemical particle properties. In terms of physical properties, the ice nucleation activity positively correlated with the particle porosity. Specifically, the wood charcoal particles showed higher porosity and water uptake capacity, consistent with the higher ice nucleation activity via PCF of the wood charcoal compared to the grass charcoal. In terms of particle chemical properties, the ice nucleation activity positively correlated with the presence of inorganic mass spectral features. More specifically, the mass spectra of the wood charcoal particles contained ions indicating
the presence of mineral components that were mostly absent in the mass spectra of the grass charcoal. This is consistent with findings of earlier studies that reported ice active particles generated from combustion processes to contain mineral components (Jahn et al., 2020; Petters et al., 2009; Grawe et al., 2018), and other studies that have reported ice nucleation activity of mineral components of other aerosol types (e.g., Archuleta et al., 2005).

Overall, we found considerable similarities of the grass and wood charcoal average mass spectra with those reported in the
literature for particles classified as "biomass burning particles" and "(coal) fly ash particles", respectively. This could indicate



that charcoal particles as studied here are atmospherically more widespread and abundant than previously thought. This is corroborated by recent findings of Adachi et al. (2022), who reported fine-ash particles (classified by the authors as particles having Ca and Mg > 0.5 wt%) to be a major constituent associated with biomass burning, estimating their global emission to range between 8.8 Tg yr$^{-1}$ to 16.3 Tg yr$^{-1}$. Taken together, these studies reveal that clear identification, differentiation and

quantification of the various particle types associated with complex biomass burning emissions remains challenging but is essential to confirm assumptions about their abundance and environmental effects made in models. Future laboratory and field measurements are crucial to the further improve our understanding of the discriminative physical and chemical (composition) properties of the particle types emitted from biomass burning, and how these in turn affect their ice nucleation activities, in particular in the cirrus temperature regime where the availability of data remains scarce.

In the present study, we avoided generalizations about the relative importance of physical over chemical particle properties in determining ice nucleation activities given the correlational nature of our ice nucleation results and the physicochemical particle properties investigated here. Furthermore, in view of INPs making up only a small fraction of the total aerosols emitted, correlations with bulk particle properties should be interpreted carefully. Therefore, for future studies it will be crucial to more holistically understanding the physical and chemical properties of the highly complex biomass burning aerosol on the level of

single particles that form and do not form ice. Such analysis could be achieved from examining ice crystal residual particles, to attribute the ice nucleation activity to their properties with more certainty and disentangle their relative contributions and importance, by comparing to properties of interstitial particles that did not form ice crystals.



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

**Figure 1**: Mean activated fraction ($AF$) of $d_m$ = 400 nm charcoal particles as a function of temperature, as determined with
IMCA-ZINC. The grey shaded region indicates homogeneous freezing conditions, as calculated using the homogeneous nucleation rates reported in Ickes et al. (2015). Error bars denote the uncertainty in the activated fraction due to particle counts of the CPC and OPC.

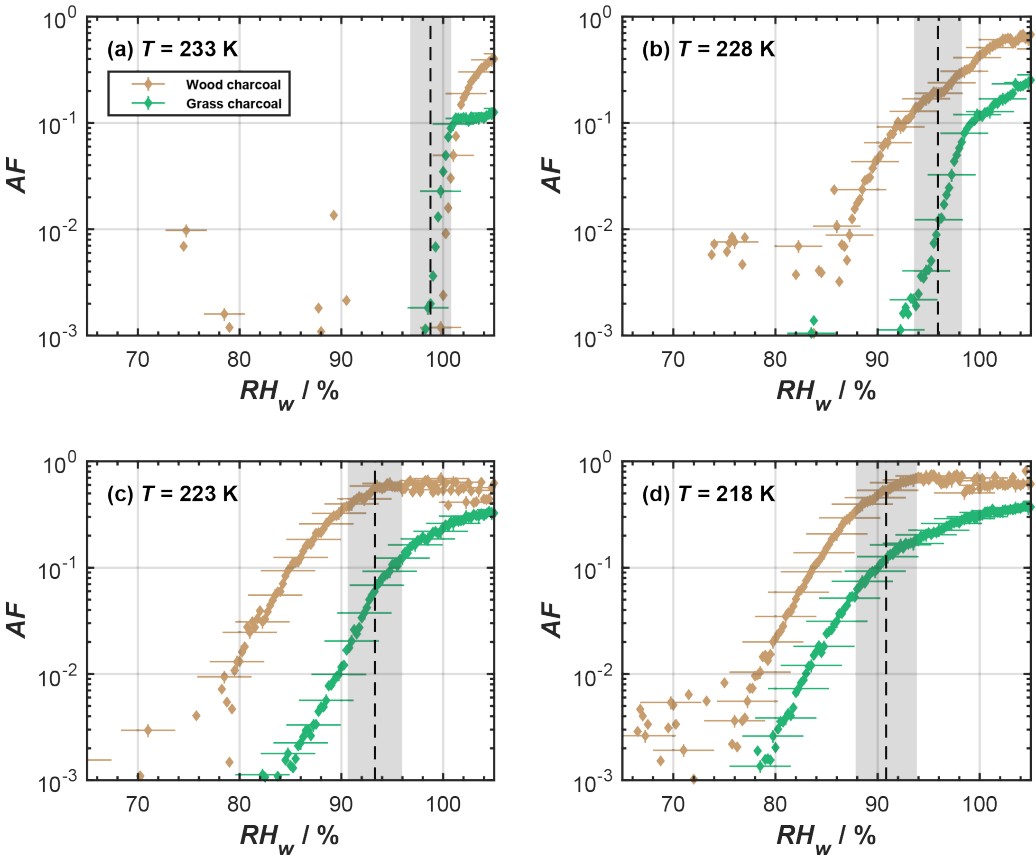

**Figure 2:** Mean activated fraction ($AF$) as a function of RH$_w$ of the different size selected charcoal particles ($d_m$ = 400 nm), as determined from relative humidity scans in HINC for cirrus regime temperatures of (a) 233 K, (b) 228 K (c) 223 K and (d) 218 K. The black, dashed lines represent expected homogeneous freezing conditions according to Koop et al. (2000), and the gray shaded regions indicate the calculated RH$_w$ variation across the aerosol lamina in HINC (Mahrt et al., 2018). Uncertainties are only given for every fifth data point for clarity.





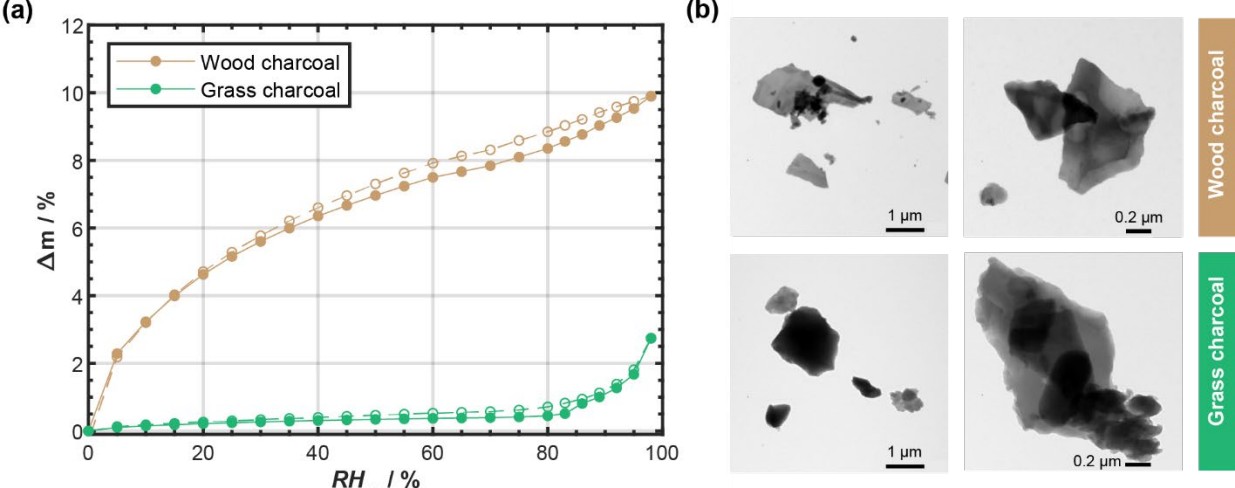

**Figure 3**: (a) Water uptake (solid lines, filled symbols) and loss (dashed lines, open symbols) isotherms given as relative sample mass change, $\Delta m$, as a function of relative humidity with respect to water ($RH_w$), as determined by dynamic vapor sorption. Sorption isotherms were measured at $T = 298$ K, and the data points represent water up take and loss at quasi-equilibrated $RH_w$ conditions. Lines are to guide the eye and uncertainty in $\Delta m$ is 0.75 %. (b) Example transmission electron microscopy images of wood charcoal and grass charcoal.

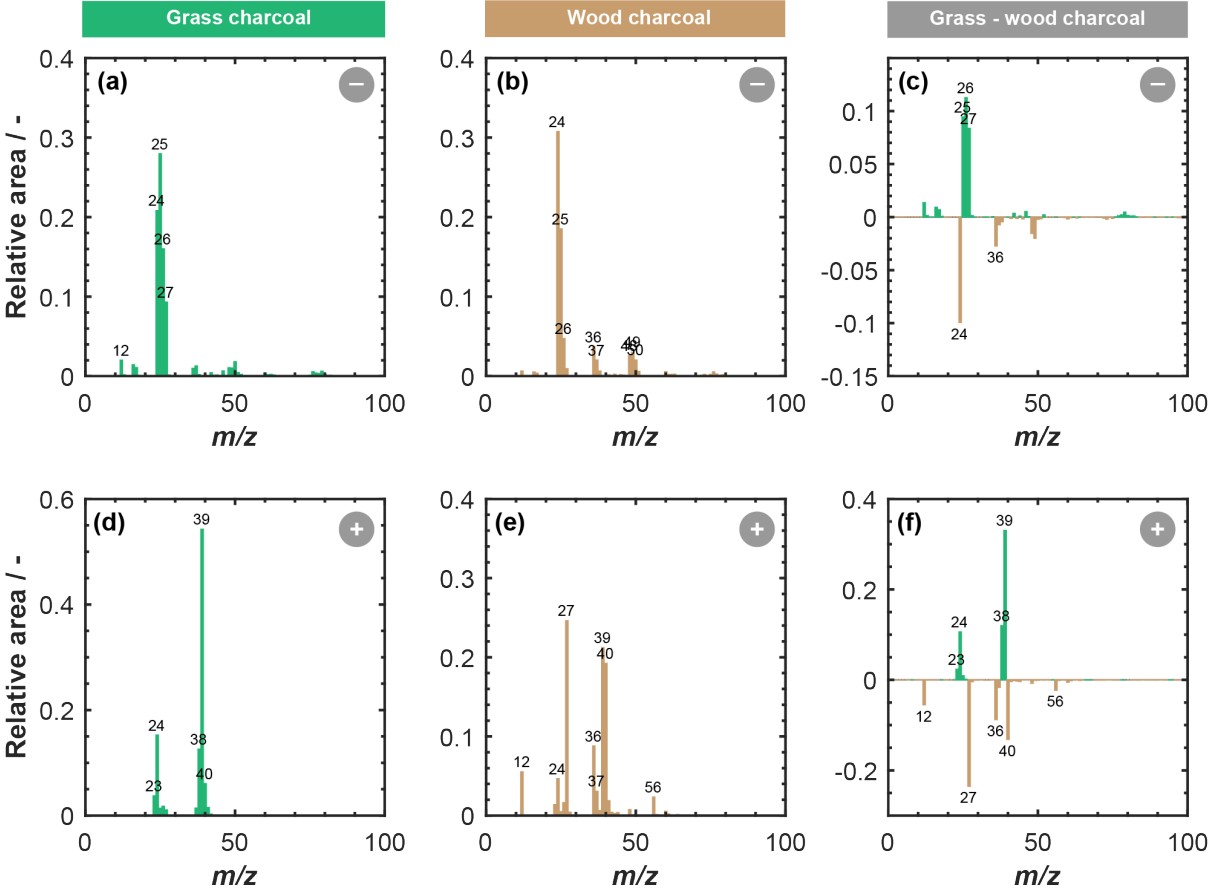

**Figure 4:** Average ATOFMS mass spectra for grass charcoal particles and wood charcoal particles for different polarities:
negative mass spectra for (a) grass charcoal and (b) wood charcoal; positive mass spectra for (d) grass charcoal and (e) wood
charcoal. The panels in the right-hand column compare the two charcoal types and show the difference in relative peak area
for the wood charcoal signal being subtracted from the grass charcoal, for anions and cations, respectively. A positive (nega-
tive) signal in the panels (c) and (f) corresponds to a larger (lower) relative peak area signal of the respective ions for the grass
charcoal compared to the wood charcoal. All spectra are shown in terms of relative peak area and for unit mass resolution.
Each single-particle spectrum was first normalized and then the average was calculated over 11'135 (grass) and 12'357 (wood)
spectra. Peaks with relative area signals larger than 0.02 are labelled with the corresponding mass to charge (*m/z*) ratio.





## Appendix A: Ice nucleation

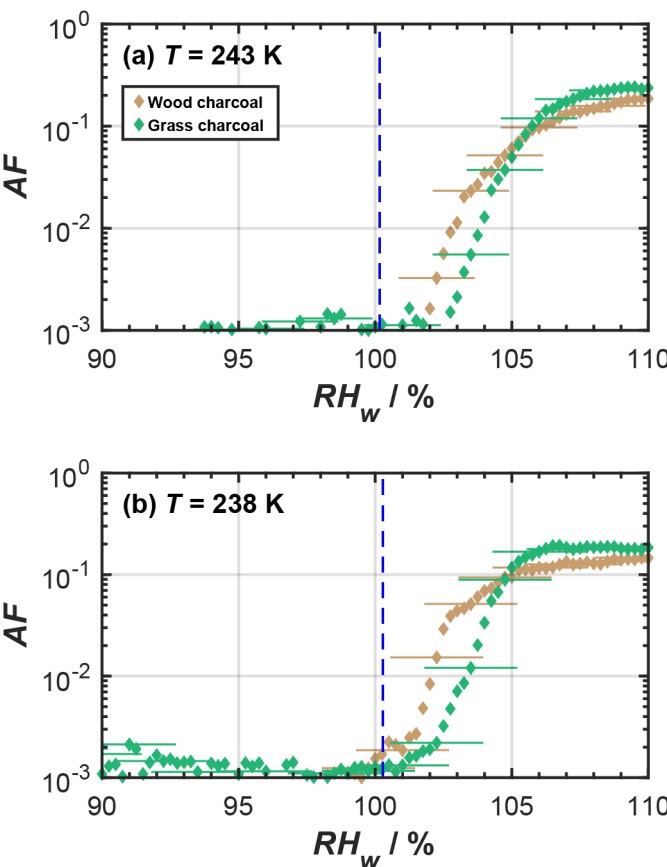

**Figure A1:** Mean activated fraction (*AF*) as a function of $RH_w$ of size selected charcoal particles ($d_m$ = 400 nm), as determined
from relative humidity scans in HINC for mixed-phase cloud temperatures of (a) 243 K, (b) 238 K. The blue, dashed lines
represent water droplet survival for a particle of initial diameter of 400 nm, a residence (growth) time of 10 s, and assuming
pure condensational growth, following Lohmann et al. (2016). Uncertainties are only given for every fifth data point for clarity.



## Appendix B: Aerosol time-of-flight mass spectrometry

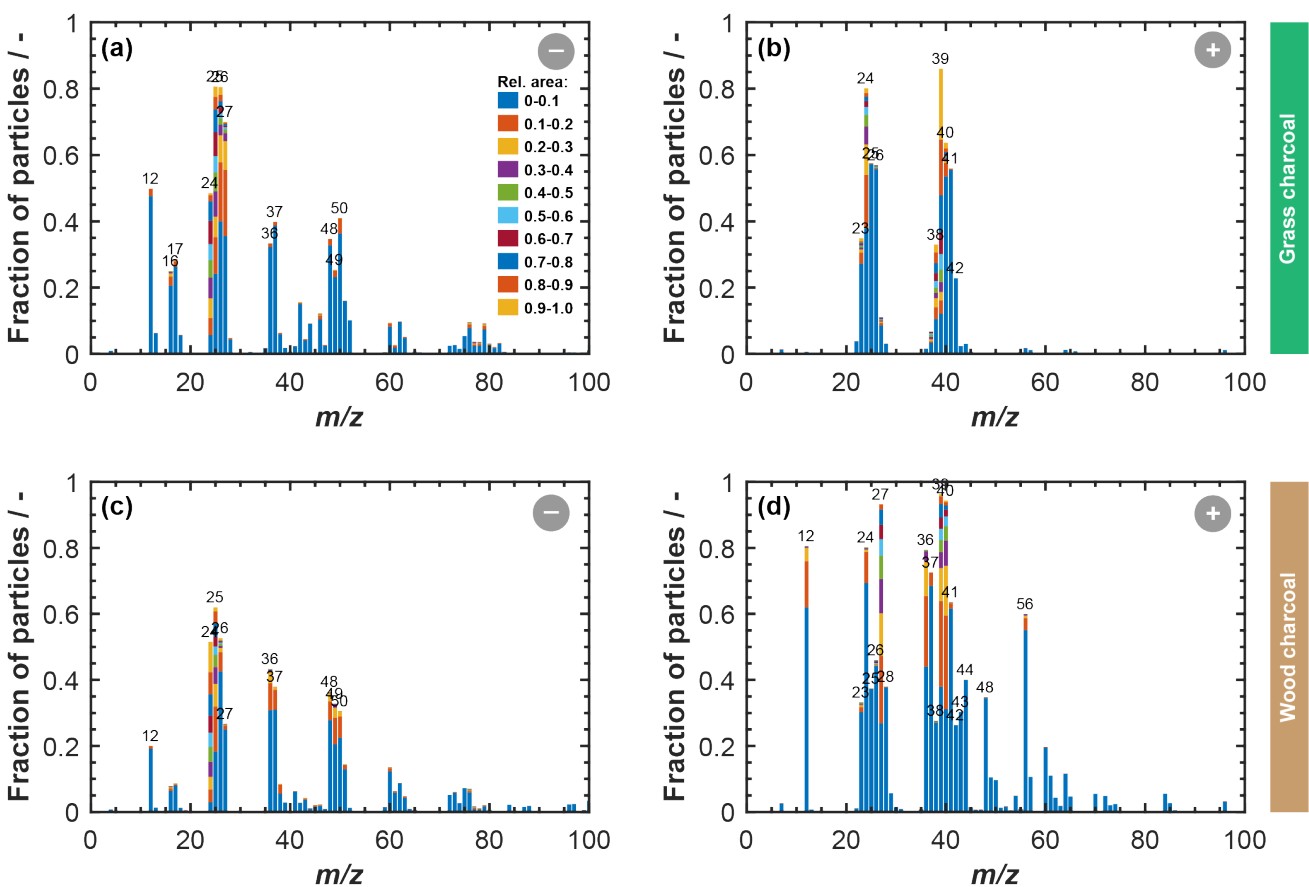

**Figure B1:** Average positive and negative ATOFMS mass spectra for grass charcoal (a, b) and for wood charcoal (c, d). In each panel the mass spectra are shown in terms of the fraction of particles out of all particles of a given charcoal type, that contain a peak with relative peak area values as indicated by the colour code.






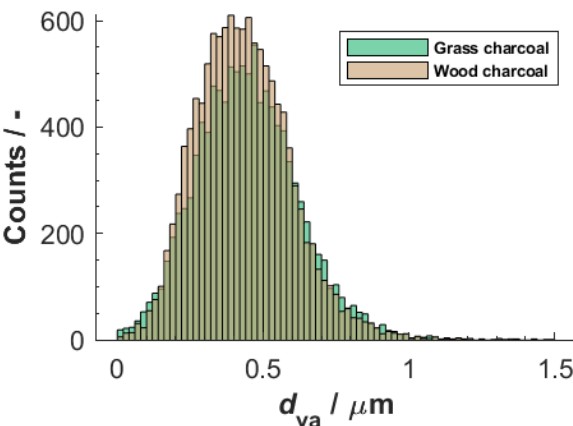


**Figure B2:** Histogram of particle size for grass and wood charcoal samples, based on the ATOFMS data, showing the vacuum aerodynamic diameter ($d_{va}$). Single particle data was binned into 20 nm bins between 0 μm < $d_{va}$ ≤ 1.5 μm.





**Appendix C: Transmission electron microscopy**


**Figure C1:** Exemplary transmission electron microscopy images of representative charcoal particles, sampled from a polydisperse aerosol population. Images show (a)-(f) grass charcoal particles and (i)-(l) wood charcoal particles, respectively.





## Appendix D: Dynamic vapor sorption

The specific particle surface area was determined from the water vapor adsorption isotherms measured by DVS and using the BET method. Specifically, the surface area values were determined from the linear BET plot of the $H_2O$ isotherm (Fig. D1) up to between $p/p_0 = 0.15$ to $p/p_0 = 0.35$, where p is the equilibrium pressure and $p_0$ denotes the saturation vapor pressure, assuming a $H_2O$ molecular cross-sectional area of $\sigma_{H2O} = 0.114$ nm$^2$, using the linear form of the BET equation Thommes et al. (2015):

$$\frac{\frac{p}{p_0}}{n(1-\frac{p}{p_0})} = \frac{C-1}{n_m C}\left(\frac{p}{p_0}\right) + \frac{1}{n_m C}. \quad (D1)$$

Here n is the total amount of water vapor adsorbed on the particles, $n_m$ is the specific monolayer capacity, and C is a constant. The water vapor-based BET surface areas were determined as 161.3 m$^2$ g$^{-1}$ and 7.9 m$^2$ g$^{-1}$ for wood charcoal and grass charcoal, respectively.


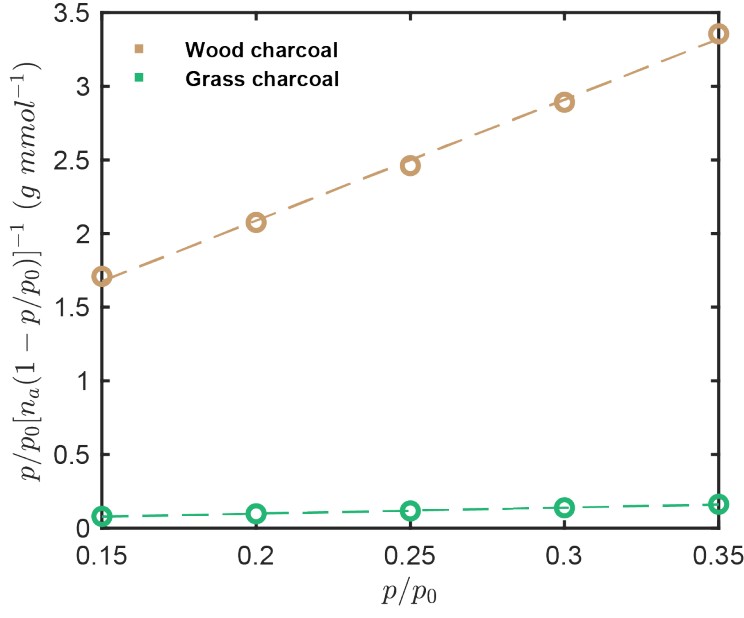

**Figure D1**: BET plot of the charcoal samples.