# Peer review of "Physicochemical properties affect ice nucleating abilities of biomass burning derived charcoal aerosols at cirrus and mixed-phase cloud conditions"

_Atmospheric Chemistry and Physics, 2022_

## Author Response (AR1)

**Dear editor,**

Thank you for your time and the assessment of our manuscript. Please find below detailed responses to each reviewer comment. We have highlighted our responses in blue font. Changes made to the text are given in blue, italic font. In addition, we have numbered the reviewer comments for ease of cross-reference.

**Thank you for your time in assessing our work.**

**Reviewer #1**

This manuscript presented ice nucleation properties of biomass burning generated charcoal particles and how the physicochemical properties affect their ability to form ice. Ice nucleation and related ice nucleating particles in the atmosphere are still poorly understood. This study measured the ice nucleation activity of charcoal particles at different temperature ranges. It provides additional new data sets and also a possible nucleation pathway for these particles. The scope of this manuscript is suitable for this journal. A few issues and comments need to be considered before publication.

We thank the reviewer for their time and for providing valuable feedback and comments to our work. Below we address each comment individually.

**Comments:**

R1C1: Line 162 and Line 212, How is temperature uncertainty of 0.1 K determined? Temperature uncertainty stated here for both of the setups might have been underestimated especially at low temperatures. The uncertainty of the temperature sensor its own could have already  $\pm$ 0.1 K? Any consideration on the temperature variations and distribution inside the flow tube in such large devise?

The temperatures of the continuous flow diffusion chamber (CFDC) walls can only be controlled within the uncertainty range of the thermocouples, which according to the manufacturer specifications have an uncertainty of 0.1 K across the temperature range the CFDC is operated in our experiments. However, this does not correspond to the temperature uncertainty that the aerosol layer experiences in the CFDC. In our experiments, the temperature of each CFDC wall was separately controlled by individual ethanol recirculation chillers, which continuously flush ethanol through the cooling pipes of the CFDC walls, having a constant temperature close the desired wall temperature. Temperature variations of the walls due to heat loss to the environment are minimized by covering each CFDC wall with a ~5 cm thick insulation layer of foamed material. This setup results in maximum temperature variations along each CFDC wall of 0.1 K as well.

By setting the CFDC walls to different sub-zero temperatures, leads to a linear temperature distribution inside the chamber, specifically across the vertical extent of the chamber. This linear temperature gradient then results in a supersaturation profile between the two walls, having a parabolic shape in the direction of flow, as described in the text.

The uncertainties in relative humidity (RH) of our ice nucleation experiments then result from two factors: The first factor is the temperature uncertainty of 0.1 K along each CFDC wall that set the boundary conditions of the linear temperature gradient and ultimately control the RH-

profile across the vertical chamber extent. The second factor is the thickness (vertical extent) of the air layer in which the particles are transported through the chamber, termed "aerosol-lamina". This thickness is controlled by the ratio of aerosol-to-sheath flow. The smaller this ratio, the thinner is the aerosol lamina and thus the smaller the RH uncertainties, which result from the variation of RH along the supersaturation profile, specifically its portion covered by the aerosol lamina.

By choosing a lower-limit of 1:10 for our aerosol-to-sheath ratio and evaluating the variation of RH across the aerosol lamina at the lowest experimental temperature of 218 K, where spread in RH across the aerosol lamina is largest, we conservatively calculate upper-limit RH-uncertainties of our experiments. This ensures that the relative humidity uncertainties are not underestimated at low temperatures, but instead are overestimates at higher temperatures. Please see Appendix D1 of Mahrt et al. (2018) for a more detailed discussion.

To clarify this in the text, we have made the following changes in Section 2.2.1 (L195):

"Briefly, in HINC two ice-coated, horizontally oriented copper plates are separately cooled to different temperatures by individual recirculating coolers, achieving a temperature control along each of the CFDC walls of  $\pm 0.1$  K, corresponding to the thermocouple uncertainty. By setting the two walls of HINC to different sub-zero temperatures, so that a linear temperature gradient is established across the vertical extent of the ice chamber.

**[...]**

The vertical extent of the aerosol lamina controls the portion of the supersaturation profile that can be experienced by the particles within HINC, i.e., it governs the variations of T and RH the particles are exposed to. The RH-conditions across the aerosol lamina at the center of the HINC chamber were calculated from the linear temperature profile between the two horizontal chamber walls, the parametrization given by Murphy and Koop (2005), and a maximum temperature uncertainty of either CFDC wall of 0.1 K. Assuming a lower limit for the aerosol-tosheath flow ratio of 1:10.this translates to an upper limit uncertainty of the relative humidity with respect to water (RHw; and with respect to ice, RHi) of approximately  $\pm 3$  % (5 %) at T = 218 K. At the same center temperature, and for a center RHw = 105 %, i.e. when the temperature gradient between the two walls is largest, the T variation across the aerosol lamina is approximately 1.1 K, again assuming an aerosol-to-sheath flow ratio of 1:10. These maximum RH- and T-uncertainties are conservative estimates and decrease for ice nucleation experiments at lower temperatures, as detailed elsewhere (e.g. Appendix D2 of Mahrt et al., 2018)."

**R1C2: Figure 1, do you have any blank measurements of pure water droplets or aqueous inorganic salt droplets?**

To address the reviewer's comment we have added reference measurements of aqueous inorganic salt droplets to our Figure 1. Upon adding the data, we realized that the homogeneous freezing conditions within our original plot followed the data by Earle et al. (2010) and not by Ickes et al. (2015), as erroneously indicated in the figure caption. We have corrected this accordingly.

R1C3: Line 182, Does the OPC measure the ice crystals? How long does it take to grow to a 1 um size ice crystal? The residence time is only 10 seconds. Is it sufficient to grow ice crystal at such low water vapor pressure and low temperatures? I would guess the OPC operates at

room temperature, it measures the size of droplets from the melting of ice crystals? How do the temperature and RH affect the size of droplets or ice crystals in the tube when they transport from the HINC to OPC? Evaporation or melting during the transportation.

The OPC simply counts particles and determines their size based on the intensity of the scattered light signal. In CFDC studies, a size threshold is commonly used to discriminate between liquid water droplets and solid ice crystals. Discrimination of particle type by (optical) size can be achieved, because of the different saturation ratios with respect to water and ice, resulting in different growth rates of water and ice particles for given conditions of temperature and relative humidity within HINC. However, given that our HINC measurements were performed at  $T \le 233$  K, i.e., below the homogeneous ice nucleation temperature of water, particles detected by the OPC in our HINC experiments must be ice, independent of their size.

Growth of the ice particles to diameters of 1  $\mu$ m or beyond within the 10 s residence time is commonly achieved within HINC. The hydrometeor growth within HINC has been discussed in detail in Mahrt et al. (2018; see their Appendix D2) for a wide range of conditions T and RH conditions. Specifically, in our previous study, we have assumed pure diffusional growth of idealized spherical ice crystals from the vapor phase together with an accommodation coefficient of  $\alpha = 0.1$  to calculate the size of ice crystals as a function of their time within HINC and their position within the aerosol lamina (see e.g. Fig. D3 in Mahrt et al. 2018). The results of similar diffusional growth equations, but updated for an initial particle radius of 200 nm, corresponding to the size of our charcoal particles studied here, and for two different accommodation coefficients of  $\alpha = 0.1$  and  $\alpha = 1.0$  (Skrotzki et al., 2013), covering a range of possible deposition coefficients, are shown in Figure 1 below. These theoretical calculations reveal that for characteristic conditions, particles often grow to diameters ~2  $\mu$ m in diameter (note that in the figure below radius instead of diameter is plotted).

**Figure 1:** Expected diffusional growth of spherical ice crystals, having an initial diameter of 200 nm, within HINC for different cirrus regime temperatures,  $T_{center}$  (rows), and  $RH_w$  along the centerline of HINC (columns), as a function of residence time. Solid lines indicate the ice crystal radius (right-hand ordinate). Dashed indicate the vertical distance covered by the ice crystal (left-hand ordinate). Calculations were performed as described in Mahrt et al. 2018 (see their Appendix D2) and stopped once the particles settled vertically by 1 cm. The values of  $\alpha$  indicate different accommodation coefficient of water molecules on ice.

The reviewer is correct that the OPC operates at room temperature. The OPC is placed immediately at the exit of the cloud chamber and is connected by a long tube of length s= 3 cm, that has an inner radius of r = 2 mm. Using a volume of the connector tube of  $V_{cylinder} = \pi \cdot r^2 \cdot s = 0.0021$ cm3, and the volume flow rate from the CFDC to the OPC of Qflow = 2.83 L min-1, yields a transit time of t =  $V_{cylinder}/Q_{flow} = 8 \cdot 10^{-5}$  s. Given the short transit time between the cloud chamber and the OPC changes in the hydrometeor size due to sublimation, melting and/or evaporation can be neglected. To further minimize changes in hydrometeor size the connector tube between the CFDC and the OPC is insulated, ensuring a temperature well below 0 °C.

To clarify these aspects, we have added the following statement to the revised manuscript (L223):

"Given that the HINC experiments were all performed at  $T \le 233$  K, i.e. below the homogeneous ice nucleation temperature of water, all particles considerably larger than that of the sizeselected aerosols particles must be ice particles. Theoretical growth calculations that have assumed pure diffusional growth of idealized spherical particles, using a conservative deposition coefficient of 0.1, suggest that ice particles form to diameters beyond 1 µm within the residence time of HINC (~ 10 s) for a wide range of temperature and relative humidity conditions, as detailed in our earlier work (Mahrt et al., 2018). Given the similar temperature and relative humidity conditions used in our experiments herein, we expect formation of ice particles sufficiently large to be detected by the OPC. Related, decrease in the ice particle size due to sublimation or melting during the transition from the exit of HINC to the OPC is assumed to be negligible, given the short transit time (< 10-4 s)."

R1C4: L284, change "immersion freezing experiments" to "freezing experiments" or "ice nucleation experiments" since you do not know if the ice nucleation is through homogeneous or heterogeneous nucleation.

**Changed to "freezing experiments"**

R1C5: L315 How do these ice nucleation onsets for charcoal particles at low temperatures compared to other biomass burning aerosols?

We believe that with "low temperatures" the reviewer refers to the cirrus temperature experiments presented here. We note that the availability of cirrus temperature ice nucleation measurements of biomass burning aerosol is very limited. One previous study by DeMott et al. (2009) has found the particles to freeze alongside homogeneous freezing of solution droplets. In this regard, the charcoal particles investigated here are much better heterogeneous INPs at  $T \leq HNT$ . We have already noted this on L90 of the original manuscript. In order to clarify this and to emphasize that more cirrus temperature ice nucleation measurements of biomass burning are needed, we have added the following statements to the manuscript (L419):

"Yet, at  $T \leq HNT$  the charcoal particles revealed heterogeneous ice nucleation at conditions well below those relevant for homogeneous freezing of solution droplets, making the charcoal particles investigated here considerably better cirrus ice nuclei compared to other biomass burning derived particles investigated earlier (DeMott et al., 2009)."

R1C6: L319 I do not follow the reasoning that the ice nucleation activity of the charcoal particles at cirrus temperature regime is result from PCF. Different ice nucleation ability of particles at different temperature ranges, immersion freezing and deposition mode nucleation, can simply because the presence of liquid water changed the active sites.

If the observed heterogeneous ice nucleation in our HINC experiments was due to classical deposition nucleation, i.e., ice formation by direct deposition from the vapor phase, there should also be some ice formation resulting from deposition nucleation at T = 233 K and RH  $< RH_{hom}$ , which is absent in our experiments (Fig. 1a). There would be no reason for deposition nucleation to abruptly kick in once the temperature is brought down to below the HNT. In addition, we would expect ice formation by deposition nucleation to proceed at T > 233 K and RHw < 100%, which is also absent in our experiments. The latter has been verified by the additional HINC experiments presented in Fig. 1A, where the AF curves only start sloping upwards above 100 % RHw, i.e. conditions where deposition nucleation is believed to be absent due to thermodynamic conditions favoring condensation of bulk water drops. Given the strong

dependence of ice nucleation on the homogeneous ice nucleation temperature of water (HNT  $\approx 235$  K), we suggest that the ice formation observed in our experiments is due to PCF, i.e., homogeneous freezing of pore/capillary water and subsequent growth of pore ice into macroscopic ice crystals. This interpretation is consistent with earlier studies. For instance, the study by David et al. used porous silica particles and showed that ice formation by PCF is present at  $T \leq 233$  K, but not at T = 238 K (see their Fig. 1). The same study also found no ice formation by deposition nucleation at T = 238 K, consistent with our results presented herein.

In short, the reasoning why we interpret the ice formation observed in our HINC experiments to result from PCF has been described in detail on L334-337 of the original manuscript. We have made the following adjustments to further clarify our reasoning in response to the referee comment (L396):

"Additional RH-scans performed with HINC at temperatures of 243 K and 238 K (see Fig. A1) confirmed the absence of ice formation below water saturation for T > HNT., Such a distinct increase in the ice formation ability at  $T \le HNT$  cannot be explained by classical nucleation theory assuming ice formation via deposition nucleation, where a liquid water phase is absent (Welti et al., 2014). Thus, the marked dependence of ice formation on HNT suggests that the charcoal particles investigated herein form ice via PCF, where ice formation proceeds via homogeneous freezing of liquid water in pores for  $RH_w < 100\%$ . These interpretations are consistent with observations of previous studies (e.g. David et al., 2019)."

R1C7: It would be helpful if the manuscript briefly describes the different types of isotherm in section 3.2 or SI. This is important to reader to gain a better understanding on the distribution of different micropore or mesopore in these charcoal particles.

To address the reviewer's comment, we have added some more details to our descriptions of the isotherm and hysteresis types associated with our charcoal particles at various instances in Sect. 3.2 of the revised manuscript. For a more in depth analysis and description of the individual isotherms we refer the reader to the work by Thommes et al. (2015), as clarified within our manuscript.

R1C8: Figure C1, what are the length of the scale bars?

Thank you for spotting this. We have increased the readability of the scale bars.

**Reviewer #2**

The authors present measurements of the ice nucleation ability (immersion and pore condensation freezing modes), water sorption profiles, and limited chemical composition measurements using a laser ablation mass spec of charcoal-derived particles from one grass and one wood charcoal sample which they resuspend. The methods used are sound, as are the conclusions made. Connecting this work to atmospheric conditions and possible effects is a bit more challenging due to the nature of the charcoal samples used and the limited chemical analysis performed. Still this work is a notable contribution to our understanding of the possible particle types emitted from biomass burning and their effects on cloud microphysics. It should be of interest to the readers of ACP and suitable for publication after some aspects are better clarified and discussed in the manuscript. We thank the reviewer for their time and effort in providing critical feedback to our manuscript, which helped us to clarify important aspects of our work. Below we address each comment individually and hope that our answers and changes made alleviate any remaining concern's the reviewer raised.

R2C1: Since pyrolysis-derived charcoal particles are studied here I do not think using "biomass burning" in the title is appropriate, however. This could be a bit misleading since combustionderived particles were not studied here and this is a very different process from pyrolysis (which is not directly relevant to actual biomass burning that emits particles into the atmosphere).

The charcoal particles were produced by pyrolysis under an N2 environment. Clearly, this will produce very different particle properties and compositions that occurs in actual wildfires and prescribed burns that proceed over a large range of combustion conditions in oxygen-containing air. There was not much discussion of how the charcoal particles studied here might relate to components of biomass-burning particles that are actually emitted to the atmosphere. The authors should really add a discussion of this so that it is clear that their objective is to focus specifically on the properties of charcoal particles that were produced through pyrolysis and not combustion processes. This is fine and provided some valuable insights, it will just make it difficult to generalize to atmospheric conditions and properties.

The reviewer raises a fair point, that the processes of combustion and pyrolysis are mechanistically different and hence the produced particles have different properties. As the reviewer points out correctly, the charcoal particles studied here were produced by pyrolysis. In order to reflect this better, and in accordance with the reviewer's suggestion, we have changed the title of the revised manuscript to:

**"Physicochemical properties of charcoal aerosols derived from biomass pyrolysis affect their ice nucleating abilities at cirrus and mixed-phase cloud conditions".**

In order to further clarify the difference between the combustion process and the pyrolysis process, and to our objective here was to study charcoal particles from pyrolysis, we have added the following statement to Section 2.1 (L153):

"Pyrolysis describes the thermochemical process where the biomass is decomposed at elevated temperatures within an environment where oxygen is limited (Lehmann and Joseph, 2015). In contrast to pyrolysis, combustion describes the thermochemical process involving the reactions of the biomass with oxygen, where heat is released (Glassman et al., 2014). Biomass pyrolysis generally leads to the formation a range of different compounds including gaseous (volatile), liquid and solid products, with the relative distribution of products being mainly dependent on the pyrolysis conditions (e.g. temperature, heating rate) and fuel material (Demirbaş and Arin, 2002; Williams and Besler, 1996). Charcoal particles encompass the solid residue that is formed in a pyrolytic atmosphere."

To further address the reviewer's comment related to the components of biomass burning particles, we have also considerably extended the discussion how different particle are emitted into the atmosphere during biomass burning, through various changes throughout Section 1. Please see changes directly within Section 1 of the revised manuscript. Lastly, in order to discuss how the charcoal particles studied here might relate to those emitted during real wildfire we have added the following statement (L173):

"The temperature of biomass burning in e.g. real wildfires occurs over a range of different temperatures (e.g., Mondal and Sukumar, 2014). Thus, in the natural environment, biomass pyrolysis at different temperatures can resulting in charcoal particles with different physicochemical properties. As such atmospheric charcoal particles can have different properties to the particles studied here."

R2C2: Along with this, what contribution organic aerosol components make to the charcoal particle studied should be better clarified. In combustion-derived BBA there is a variable and significant amount of organic aerosol material that spans a range of volatility down to quite low saturation vapor pressures. This often includes tar like material that produces tar ball particles that can also mix with and coat other BBA components. I imagine these organics could also fill or conceal the pores needed for pore condensation freezing. Since pyrolysis and not combustion-produced particles were studied here, and a N2 purge gas was used, it seems that there is little contribution from such organic aerosol components here? This is important to clarify, and again makes the charcoal particle studied here rather poor mimics of actual combustion-derived BBA.

We agree with the reviewer that during biomass pyrolysis organic material of variable volatilities and in particular tar-like material is often formed. The reviewer is further correct that such material can block pores and render these inaccessible for ice nucleation via PCF. To clarify this we have added the following statement (L594):

"In this regard we recognize that our results are confined to charcoal particles produced by low temperature pyrolysis, and hence cannot necessarily be generalized to atmospheric charcoal particles and other particles emitted by biomass burning. Certainly, in real biomass burning events the temperature can vary considerably (Mondal and Sukumar, 2014), affecting pyrolysis yields and particulate and gas product distributions (Demirbas, 2007; Safdari et al., 2019). For instance, Safdari et al. (2019) have reported tar-like material to make up between 44 % to 62 % by weight of the particulate products when investigating the pyrolysis of 14 common plant species native to the United States, consistent with other work that has reported tar particles to be a dominant component of biomass burning emissions (e.g., Adachi et al., 2019). Interaction of charcoal particles with particulate and gas components co-emitted during biomass pyrolysis can alter their ice nucleation abilities. For example, tar-like material and condensing organic vapors can fill pores available on the charcoal particles and hence render them inaccessibly for PCF ice nucleation, an effect that has previously been observed for soot particles (e.g., Gao et al., 2022; Zhang et al., 2020)."

The reviewer is also correct that such material has been removed by the nitrogen purge flow during the generation of the charcoal particles studied here (L159).

**"Any organic material associated with the biomass fuel that is volatile at the pyrolysis temperature was removed from the sample by the purge flow during the particle preparation."**

R2C3: Related, Jahl et al. (2021) reported the enhancement of INA in BBA from grass fuels through simulated atmospheric aging. This was attributed to the removal of organic coatings that concealed the mineral ice active surface sites. I imagine similar processes likely occur that

alter the availability or properties of the pores involved in PCF also. While the pyrolysis charcoal particles studied here are too simplistic to provide this sort of behavior, this is why it is important to discuss the contribution of organic carbon in these particles, and what role this might play in the artificial charcoal particle studied here, and realistic combustion-derived BBA.

Thank you for pointing this out. We agree with the reviewer that atmospheric aging processes can also lead to an enhancement in ice nucleation activity. Certainly, if ambient, mesoporous charcoal particles are coated with organic material, dilution driven evaporation of the organic material or other aging processes could lead to a removal of the organic material and an enhancement of the charcoal ice nucleation activity, similar to the observations of Jahl et al. (2021). To address this, we have added the following statement (L604):

"Related, it was recently reported that biomass burning particles can undergo atmospheric aging processes, leading to removal of organic coatings that mask the aerosol's ice-active sites, ultimately enhancing their ice nucleation activity (Jahl et al., 2021). Such effects were absent for the charcoal particles studied here, as the organic vapors formed during pyrolysis were continuously removed in the experimental setup used do generate the particles, as described above."

The reviewer is also right that in the real environment charcoal particles will be associated with organic material. As such, we fully agree that the organic material associated with atmospheric charcoal particles and other biomass burning aerosols needs to be investigated more closely, to better understand its impact on the ice nucleation activities of these particles. Yet, potential changes resulting from atmospheric aging processes, as brought up by the reviewer, are an important topic to study. Therefore, we have added the following statement (L610):

"In particular it will be important to better understand the role of organic material associated with atmospherically realistic charcoal particles in determining their ice nucleation activity, and how this is impacted by atmospheric aging processes."

R2C4: In the ATOFMS results I did not understand why m/z + 26 and +27 were attributed to markers/fragments of nitrogen oxides. You often do get an NO+ fragment at m/z + 30. What do the authors propose are the ions at +26 and +27?

The reviewer is right that NO fragments often occur at m/z 30. We do not, however, observe such an ion peak in the average anion mass spectra of our grass charcoal sample. We thus had originally attributed the peaks at m/z 26 and 27 to CN- and NCH- ions, respectively. However, based on the reviewer's comment, we critically re-assessed the literature and revised our statement as follows (L491):

"The presence of a peak at 46 (NO2-), even though at very low relative intensity, renders the presence of nitrogen containing ions at m/z of 26 (CN-), and 27 (NCH-) possible, further supported by the peak at 42 (CNO-). Ion peaks at 26 (CN-) and 42 (CNO-) have previously been used as an indicator of biological aerosol (Sierau et al., 2014; Creamean et al., 2013; Pratt et al., 2009), which should be absent in the charcoal particles used here. Thus, the anion peaks at m/z of 26 and 27 in our average grass charcoal spectra, are more likely attributable to hydrocarbons (C2H2- and C2H3-, respectively), or less likely but still possible, organic nitrogen (CN- and HCN-), consistent with these peaks being previously been observed for cellulose particles and particles from biomass burning (Schmidt et al., 2017)."

R2C5: It is not clear what the purpose of the ATOFMS analysis is. What about the particles is being learned here that informs their ice nucleation or water uptake properties? This section did not add much to the paper as it is currently presented.

The authors could also try to apply the OC/EC mass ratio estimates that have been demonstrated using laser ablation mass spec analysis, e.g.:

Spencer, M. T., & Prather, K. A. (2006). Using ATOFMS to determine OC/EC mass fractions in particles. *Aerosol Science and Technology*, 40(8), 585–594.

https://doi.org/10.1080/02786820600729138

Ahern, A. T., Subramanian, R., Saliba, G., Lipsky, E. M., Donahue, N. M., & Sullivan, R. C. (2016). Effect of secondary organic aerosol coating thickness on the real-time detection and characterization of biomass-burning soot by two particle mass spectrometers. *Atmospheric Measurement Techniques*, *9*(12), 6117–6137. https://doi.org/10.5194/amt-9-6117-2016

The goal of our ATOFMS analysis was to obtain information on the average chemical composition of the aerosol particles and check for systematic differences between the grass charcoal and wood charcoal particle types that could help to elucidate differences in their ice nucleation activities, as well as to assess the particle-to-particle variability within a given charcoal type. To clarify this, we have added the following statement (L314)

"The objective of our ATOFMS analysis was to gain information on the average chemical composition of the two charcoal types studied here that could help to elucidate their ice nucleation activities. In addition, by employing a single-particle mass spectrometer, the degree of heterogeneity in chemical composition within a given charcoal type (aerosol population) can be assessed. This is important for complex aerosols such as biomass burning particles, which often contain variable amounts of organic carbon, elemental carbon and inorganic material."

And on L515:

"Thus, we conclude that the variability in chemical composition within a given charcoal types is sufficiently low and that that the average spectra shown in Fig. 4 provide a reasonable picture of the composition of the particles investigated here."

A major observation of our ATOFMS results is the presence and absence of ions characteristic for mineral components in the average wood charcoal and grass charcoal spectra, respectively. We have clarified this through various changes in Sect. 3.3. Please see our answer to R2C6 below.

Lastly, we thank the reviewer for bringing these ATOFMS studies to our attention, which will be helpful for our future work.

R2C6: It is a bit odd that when discussing the ions likely derived from mineral species that the authors did not draw a connection to the recent idea that combustion-derived minerals are the main source of ice nucleants in some types of biomass-burning aerosol first presented by Jahn et al. (2020) and then supported by the field measurements of Adachi et al. (2022).

We agree with the reviewer that the presence of ions from mineral species observed in our ATOFMS analysis is in-line with earlier findings of mineral-containing species promoting ice nucleation. We also agree that these ions could stem from (pyrolysis) combustion-derived minerals, and could be a major driver of the ice nucleation activity of biomass burning particles,

as proposed by Jahn et al. (2020). As such our results are in-line with those of Jahn et al. (2020) and further support their idea. In fact, it was our intention to connect our results to these previous findings, as we have attempted with various statements throughout or work, including L103-110, L437-440, L475-477, L481-484 of the original manuscript. Based on the reviewer's comment we tried to improve our discussion on this aspect. Therefore, and to further emphasize the purpose of our ATOFMS analysis in response to R2C5, we have added the following statement to our Section 3.3 (L544):

"Similarly, recent findings have demonstrated inorganics (Ca, Mg) to be abundant in ash particles from biomass burning (Adachi et al., 2022).

Overall, our ATOFMS results reveal the presence of mineral components in the wood charcoal particles. The availability of mineral ions in the wood charcoal correlates positively with their enhanced ice nucleation activity compared to the grass charcoal particles, where ion signatures characteristic for mineral components were largely absent in the average mass spectra. These results parallel the recent findings by Jahn et al. (2020) that combustion-derived minerals, resulting from transformation of inorganic material that is naturally present within the fuel material, might govern the ice nucleation activity of particles emitted by biomass burning. This is further supported by other studies that suggested inorganic or mineral components to play a key role for the ice nucleation activity of particles emitted by biomass burning (Petters et al., 2009; Jahl et al., 2021; McCluskey et al., 2014). At the same time, we acknowledge that while our ATOFMS analysis provides valuable information on the chemical composition of the charcoal particles, including their refractory components, a detailed chemical mechanism by which these minerals promote their ice nucleation activity cannot be explicitly derived here, given the correlational nature of our data. Related, the presence of some cirrus ice nucleation activity for the grass charcoal particles, where mineral signatures were largely absent, underscores the need for future studies to better understand the relative importance of pores and active (mineral) sites for the ice nucleation activity of charcoal particles and biomass burning particles in general."

R2C7: The discussion of the lower INA observed for these pyrolysis charcoal particles compared to some other types of BBA reported such as by Jahn et al. was confusing. The authors propose that this is because of lower particle surface area in the droplets studied. But they use the widely-used n\_s metric that normalizes to surface area, so such differences should be largely normalized for. It seems far more likely that as these pyrolysis charcoal particles were produced through a very different process that combustion derived-BBA is that the resulting particles and their INA/n\_s are just very different. Jahn et al. (and Adachi et al.) attributed the higher INA observed in their BBA to mineral-containing particles. The pyrolysis method may not have produced the right conditions for volatilization and then recondensation of much of the inorganic/mineral components. Charcoal particles produced by pyrolysis will be different in many important ways from combustion particles. This is the aspect of this work that I find is the least well justified. It seems much more likely that a lower INA/n\_s was observed for these charcoal particles simply because they are entirely different from combustion BBA studied in the other reports. This is why I think the manuscript title should be changed as biomass burning particles were not actually studied here.

The reviewer is correct that the ice active surface site density  $(n_s)$  is a metric that normalizes the ice activity (activated fraction, as reported herein) to the particle surface area. However, this metric strictly only works when the particle composition is homogeneous across sizes. As such this should be kept in mind when comparing to other studies using different particle sizes where the composition may change as a function of particle size. Please note that by calculating ns "differences in particle surface area are normalized to", ultimately allowing to explore (potential) difference in ice nucleation activities across samples/studies. Such a comparison of the ice nucleation observed in our charcoal experiments to that observed previously by Jahn et al. (2020) for biomass burning particles is exactly the intention behind estimating ns values for our experiments. When taking differences in surface area between our experiments and those of Jahn et al. (2020) into account, we arrive at a maximum ns value of 80 cm-2 in our experiments compared to a maximum ns value of 8000 cm-2 in the experiments of Jahn et al. (2020) (at T =244 K). Thus, the surface area available to promote heterogeneous ice nucleation is 2 orders of magnitude lower in our case. The lower surface area available in our experiments likely contributes to the lower immersion freezing activity in our experiments compared to that observed in previous studies. We have tried to clarify this through various adjustments of the text, in order to make this comparison less confusing. The idea that an atmospherically relevant amount of surface area per droplet is an important parameter for evaluating ice nucleation, since the presence of large particles and large surface areas in a chemically inhomogeneous sample can trigger ice nucleation that would be otherwise absent in aerosol particles in the submicron range present in cloud droplets.

Please see our changes directly within Section 3.1 of the revised manuscript.

The reviewer is correct that charcoal particles produced by pyrolysis could have different properties and hence ice nucleation activities compared to combustion derived biomass burning particles investigated in other studies. To address this, we have added the following statement (L369):

"In addition, we acknowledge that different physiochemical properties of the pyrolysis derived charcoal particle types studied here compared to combustion derived biomass burning particles investigated in other studies can further contribute to differences in observed ice nucleation activities."

To further emphasize possible differences between the pyrolysis derived charcoal particles studied here and other particulate matter emitted during biomass burning, we have also added the following (L612):

"Future studies are also needed, to investigate the relative importance of the ice nucleation ability of pyrolysis derived char-coal to that of other particle types emitted during biomass burning."

Lastly, in order to clarify that the charcoal particles studied here were generated by pyrolysis in an oxygen-free atmosphere, we have adjusted the title of our manuscript; please see our answer to R2C1 above.

**References**

Adachi, K., Sedlacek, A. J., Kleinman, L., Springston, S. R., Wang, J., Chand, D., Hubbe, J. M., Shilling, J. E., Onasch, T. B., Kinase, T., Sakata, K., Takahashi, Y., and Buseck, P. R.: Spherical tarball particles form through rapid chemical and physical changes of organic matter in biomass-burning smoke, Proc. Natl. Acad. Sci., 116, 19336–19341, https://doi.org/10.1073/pnas.1900129116, 2019.

Adachi, K., Dibb, J. E., Scheuer, E., Katich, J. M., Schwarz, J. P., Perring, A. E., Mediavilla, B., Guo, H., Campuzano-Jost, P., Jimenez, J. L., Crawford, J., Soja, A. J., Oshima, N., Kajino, M., Kinase, T., Kleinman, L., Sedlacek III, A. J., Yokelson, R. J., and Buseck, P. R.: Fine Ash-Bearing Particles as a Major Aerosol Component in Biomass Burning Smoke, J. Geophys. Res. Atmospheres, 127, e2021JD035657, https://doi.org/10.1029/2021JD035657, 2022.

Creamean, J. M., Suski, K. J., Rosenfeld, D., Cazorla, A., DeMott, P. J., Sullivan, R. C., White, A. B., Ralph, F. M., Minnis, P., Comstock, J. M., Tomlinson, J. M., and Prather, K. A.: Dust and Biological Aerosols from the Sahara and Asia Influence Precipitation in the Western U.S, Science, 339, 1572–1578, https://doi.org/10.1126/science.1227279, 2013.

David, R. O., Marcolli, C., Fahrni, J., Qiu, Y. Q., Sirkin, Y. A. P., Molinero, V., Mahrt, F., Bruhwiler, D., Lohmann, U., and Kanji, Z. A.: Pore condensation and freezing is responsible for ice formation below water saturation for porous particles, Proc. Natl. Acad. Sci. U. S. A., 116, 8184–8189, https://doi.org/10.1073/pnas.1813647116, 2019.

Demirbas, A.: Effect of Temperature on Pyrolysis Products from Biomass, Energy Sources Part Recovery Util. Environ. Eff., 29, 329–336, https://doi.org/10.1080/009083190965794, 2007.

Demirbaş, A. and Arin, G.: An overview of biomass pyrolysis, Energy Sources, 24, 471–482, https://doi.org/10.1080/00908310252889979, 2002.

DeMott, P. J., Petters, M. D., Prenni, A. J., Carrico, C. M., Kreidenweis, S. M., Collett, J. L., and Moosmuller, H.: Ice nucleation behavior of biomass combustion particles at cirrus temperatures, J. Geophys. Res.-Atmospheres, 114, D16205, https://doi.org/10.1029/2009jd012036, 2009.

Earle, M. E., Kuhn, T., Khalizov, A. F., and Sloan, J. J.: Volume nucleation rates for homogeneous freezing in supercooled water microdroplets: results from a combined experimental and modelling approach, Atmospheric Chem. Phys., 10, 7945–7961, https://doi.org/10.5194/acp-10-7945-2010, 2010.

Gao, K., Koch, H.-C., Zhou, C.-W., and Kanji, Z. A.: The dependence of soot particle ice nucleation ability on its volatile content, Environ. Sci. Process. Impacts, 24, 2043–2069, https://doi.org/10.1039/D2EM00158F, 2022.

Glassman, I., Yetter, R. A., and Glumac, N. G.: Combustion, 5th edition., Elsevier, Amsterdam, 2014.

Ickes, L., Welti, A., Hoose, C., and Lohmann, U.: Classical nucleation theory of homogeneous freezing of water: thermodynamic and kinetic parameters, Phys. Chem. Chem. Phys., 17, 5514–5537, https://doi.org/10.1039/C4CP04184D, 2015.

Jahl, L. G., Brubaker, T. A., Polen, M. J., Jahn, L. G., Cain, K. P., Bowers, B. B., Fahy, W. D., Graves, S., and Sullivan, R. C.: Atmospheric aging enhances the ice nucleation ability of biomass-burning aerosol, Sci. Adv., 7, eabd3440, https://doi.org/10.1126/sciadv.abd3440, 2021.

Lehmann, J. and Joseph, S. (Eds.): Biochar for Environmental Management: Science, Technology and Implementation, 2nd ed., Routledge, London, 976 pp., https://doi.org/10.4324/9780203762264, 2015.

Mahrt, F., Marcolli, C., David, R. O., Grönquist, P., Barthazy Meier, E. J., Lohmann, U., and Kanji, Z. A.: Ice nucleation abilities of soot particles determined with the Horizontal Ice Nucleation Chamber, Atmos Chem Phys, 18, 13363–13392, https://doi.org/10.5194/acp-18-13363-2018, 2018.

McCluskey, C. S., DeMott, P. J., Prenni, A. J., Levin, E. J. T., McMeeking, G. R., Sullivan, A. P., Hill, T. C. J., Nakao, S., Carrico, C. M., and Kreidenweis, S. M.: Characteristics of atmospheric ice nucleating particles associated with biomass burning in the US: Prescribed burns and wildfires, J. Geophys. Res. Atmospheres, 119, 10458–10470, https://doi.org/10.1002/2014JD021980, 2014.

Mondal, N. and Sukumar, R.: Fire and soil temperatures during controlled burns in seasonally dry tropical forests of southern India, Curr. Sci., 107, 1590–1594, 2014.

Murphy, D. M. and Koop, T.: Review of the vapour pressures of ice and supercooled water for atmospheric applications, Q. J. R. Meteorol. Soc., 131, 1539–1565, https://doi.org/10.1256/qj.04.94, 2005.

Petters, M. D., Parsons, M. T., Prenni, A. J., DeMott, P. J., Kreidenweis, S. M., Carrico, C. M., Sullivan, A. P., McMeeking, G. R., Levin, E., Wold, C. E., Collett, J. L., and Moosmuller, H.: Ice nuclei emissions from biomass burning, J. Geophys. Res.-Atmospheres, 114, https://doi.org/10.1029/2008jd011532, 2009.

Pratt, K. A., DeMott, P. J., French, J. R., Wang, Z., Westphal, D. L., Heymsfield, A. J., Twohy, C. H., Prenni, A. J., and Prather, K. A.: In situ detection of biological particles in cloud ice-crystals, Nat. Geosci., 2, 397–400, https://doi.org/10.1038/ngeo521, 2009.

Safdari, M.-S., Amini, E., Weise, D. R., and Fletcher, T. H.: Heating rate and temperature effects on pyrolysis products from live wildland fuels, Fuel, 242, 295–304, https://doi.org/10.1016/j.fuel.2019.01.040, 2019.

Schmidt, S., Schneider, J., Klimach, T., Mertes, S., Schenk, L. P., Kupiszewski, P., Curtius, J., and Borrmann, S.: Online single particle analysis of ice particle residuals from mountaintop mixed-phase clouds using laboratory derived particle type assignment, Atmospheric Chem. Phys., 17, 575–594, https://doi.org/10.5194/acp-17-575-2017, 2017.

Sierau, B., Chang, R. Y. W., Leck, C., Paatero, J., and Lohmann, U.: Single-particle characterization of the high-Arctic summertime aerosol, Atmos Chem Phys, 14, 7409–7430, https://doi.org/10.5194/acp-14-7409-2014, 2014.

Skrotzki, J., Connolly, P., Schnaiter, M., Saathoff, H., Möhler, O., Wagner, R., Niemand, M., Ebert, V., and Leisner, T.: The accommodation coefficient of water molecules on ice – cirrus

cloud studies at the AIDA simulation chamber, Atmos Chem Phys, 13, 4451–4466, https://doi.org/10.5194/acp-13-4451-2013, 2013.

Thommes, M., Kaneko, K., Neimark, A. V., Olivier, J. P., Rodriguez-Reinoso, F., Rouquerol, J., and Sing, K. S. W.: Physisorption of gases, with special reference to the evaluation of surface area and pore size distribution (IUPAC Technical Report), Pure Appl. Chem., 87, 1051–1069, https://doi.org/10.1515/pac-2014-1117, 2015.

Welti, A., Kanji, Z. A., Lüönd, F., Stetzer, O., and Lohmann, U.: Exploring the Mechanisms of Ice Nucleation on Kaolinite: From Deposition Nucleation to Condensation Freezing, J. Atmospheric Sci., 71, 16–36, https://doi.org/10.1175/jas-d-12-0252.1, 2014.

Williams, P. T. and Besler, S.: The influence of temperature and heating rate on the slow pyrolysis of biomass, Renew. Energy, 7, 233–250, https://doi.org/10.1016/0960-1481(96)00006-7, 1996.

Zhang, C., Zhang, Y., Wolf, M. J., Nichman, L., Shen, C., Onasch, T. B., Chen, L., and Cziczo, D. J.: The effects of morphology, mobility size, and secondary organic aerosol (SOA) material coating on the ice nucleation activity of black carbon in the cirrus regime, Atmospheric Chem. Phys., 20, 13957–13984, https://doi.org/10.5194/acp-20-13957-2020, 2020.